# Three Decades of REDOR in Protein Science: A Solid-State NMR Technique for Distance Measurement and Spectral Editing

**DOI:** 10.3390/ijms241713637

**Published:** 2023-09-04

**Authors:** Orsolya Toke

**Affiliations:** Laboratory for NMR Spectroscopy, Structural Research Centre, Research Centre for Natural Sciences, 2 Magyar tudósok körútja, H-1117 Budapest, Hungary; toke.orsolya@ttk.hu; Tel.: +36-1-382-6575

**Keywords:** solid-state NMR, REDOR, dipolar coupling, distance measurement, protein structure, membrane proteins, protein–ligand interactions, amyloids, cell walls, drug binding

## Abstract

Solid-state NMR (ss-NMR) is a powerful tool to investigate noncrystallizable, poorly soluble molecular systems, such as membrane proteins, amyloids, and cell walls, in environments that closely resemble their physical sites of action. Rotational-echo double resonance (REDOR) is an ss-NMR methodology, which by reintroducing heteronuclear dipolar coupling under magic angle spinning conditions provides intramolecular and intermolecular distance restraints at the atomic level. In addition, REDOR can be exploited as a selection tool to filter spectra based on dipolar couplings. Used extensively as a spectroscopic ruler between isolated spins in site-specifically labeled systems and more recently as a building block in multidimensional ss-NMR pulse sequences allowing the simultaneous measurement of multiple distances, REDOR yields atomic-scale information on the structure and interaction of proteins. By extending REDOR to the determination of ^1^H–X dipolar couplings in recent years, the limit of measurable distances has reached ~15–20 Å, making it an attractive method of choice for the study of complex biomolecular assemblies. Following a methodological introduction including the most recent implementations, examples are discussed to illustrate the versatility of REDOR in the study of biological systems.

## 1. Introduction

Solid-state NMR (ss-NMR) spectroscopy [1,2,3,4,5] has the capability of providing atomic-level structural information in poorly crystallizable, poorly soluble molecular systems not amenable to X-ray diffraction and solution NMR techniques. Methodological advances in the past decades have made it a method of choice for the study of molecular interactions and recognition mechanisms in diverse macromolecular systems including membrane proteins, cell walls, and other complex biomolecular assemblies. Unlike in the solution state, where molecules undergo fast rotational tumbling, solid-state NMR is challenged by orientation-dependent spin interactions such as chemical shift anisotropy (CSA) and dipolar coupling, leading to a broad superposition of resonances and the loss of spectral resolution. There are three main approaches to solve this problem. One approach relies on orienting the sample, either mechanically [6] or with an external magnetic field [7,8] upon which the uniaxial orientation provides a mechanism for line narrowing, yet retaining the structural information inherent in the anisotropic spin interactions. Another approach is called “multiple pulse line-narrowing” [9,10,11], which involves the rotation of nuclear spins by radiofrequency (*rf*) pulses in such a way that over a cycle the average homonuclear dipolar interaction vanishes. Finally, the third approach is magic angle spinning (MAS) [12,13] involving a mechanical rotation of the sample about an axis aligned at 54.7° (the “magic angle” at which 3 cos^2^*θ* − 1 = 0) relative to the external magnetic field. During spinning, the spatial orientations of the vectors connecting the spins are constantly changing, making their resonance frequencies time dependent. As a result, CSA averages to the isotropic chemical shift yielding a high-resolution spectrum. A shortcoming is that at a spinning speed that is fast compared with the strength of the dipolar interaction, the latter averages out to zero as well, losing valuable structural information. To overcome this difficulty, a number of pulse sequences (e.g., rotational resonance (R^2^) [14], radio-frequency-driven recoupling (RFDR) [15], and rotational-echo double resonance (REDOR) [16,17]) have been developed to reintroduce dipolar couplings under MAS. This paper focuses on the robust and versatile capabilities of REDOR, which recovers heteronuclear dipolar couplings between nuclei such as ^13^C–^15^N, ^13^C–^31^P, ^13^C–^19^F, and in more recent applications ^1^H–X as well. Since its introduction in 1989 by Gullion and Schaefer, REDOR has been implemented to investigate the composition, structure, and dynamics in a wide range of macromolecular systems including lipid-embedded membrane peptides/proteins, enzyme-inhibitor complexes, protein assemblies (amyloids and viruses), and cell walls [18,19] in laboratories around the world. Following a methodological introduction including the most recent variations on REDOR, examples are discussed to highlight the advantages and limitations together with considerations in experimental design. ***The paper is dedicated to the memory of Professor Jacob Schaefer (1938–2022)***, a true scholar and inspiring mentor, who by pioneering REDOR and combining it with inventive isotopic labeling strategies had a determining role in biomolecular ss-NMR spectroscopy.

## 2. The REDOR Measurement

The dipole–dipole coupling between two heteronuclei within a magnetic field depends upon both spatial and spin coordinates as described by the truncated dipolar Hamiltonian given in Equation (1):(1)$${\hat{H}}_{D}=\omega_{D}{\hat{I}}_{z}{\hat{S}}_{z}\;,\;\;\omega_{D}=D\;\frac{3\;{cos}^{2}\theta -1}{2},\;\;D=\frac{\mu_{o}\gamma_{I}\gamma_{S}\hbar }{8\;\pi^{2}r^{3}}$$
where *r* is the internuclear vector between the interacting *I* and *S* spins, *θ* is the angle between the I–S internuclear vector and the applied magnetic field, *γ_I_* and *γ_S_* are the gyromagnetic ratios of the *I* and *S* nuclei, *μ_o_* is the permeability of vacuum, and *ħ* is the reduced Planck constant. Accordingly, the dipolar coupling strength, *D*, is proportional to the inverse third power of the internuclear distance. MAS averages over the spatial coordinates of the Hamiltonian suppressing the dipolar interaction in a coherent manner. In the REDOR experiment, rotor-synchronized *rf* pulses operate on the spin coordinates, imposing another coherent process of comparable frequency to prevent the complete suppression of the dipolar coupling by MAS.

The REDOR experiment is performed in two parts (Figure 1A). Both parts start with the establishment of transverse magnetization of the observed spin by cross-polarization transfer [20,21,22] from the protons. In the first part, rotor-synchronized 180° pulses are applied every half rotor period on spin *I* to flip the sign of the dipolar coupling. This results in a dephasing of the transverse *S* magnetization yielding a rotational echo of diminished intensity (*S*). In the second part, without the application of 180° pulses on spin *I*, a full echo (*S_o_*) is acquired. The amount of signal attenuation is conveniently expressed as the ratio of the difference in signal intensity in the two parts (∆*S* = *S_o_* − *S*) and the full echo spectrum, *S_o_*. The extent of dephasing depends on the *I*–*S* dipolar coupling and hence on the *I*–*S* internuclear distance and the relative orientation of the *I*–*S* internuclear vector with respect to the rotor axis, as well as the dipolar evolution time (N_r_ ∗ τ_r_), where N_r_ is the number of rotor cycles and τ_r_ is the length of one rotor period. The detection of weak couplings requires longer evolution times, which can be accomplished by increasing the number of rotor cycles and/or decreasing the spinning speed. Calculated plots of dipolar evolution for selected *I*–*S* spin pairs at specific distances are illustrated in Figure 2A. A representation of the distances corresponding to two specific dipolar couplings for various spin pairs is depicted in Figure 2B.

Unlike in conventional solution NMR determination of protein structure, where ^1^H–^1^H NOESY (nuclear Overhauser effect spectroscopy) cross-peak volumes at a particular mixing time are translated into qualitative restraints such as a distance range, the analysis of dephasing curves as a function of dipolar evolution time in REDOR measurements provides a more exact measure of the interatomic distance between the recoupled nuclei, which, given the necessary care in experimental design and setup (cf. below), can reach an accuracy of ~0.1–0.2 Å. (We note that recently developed exact NOE (eNOE) approaches relying on the extraction of exact NOE rate constants using an iterative protocol between theory and experiment have been shown to reach a similar high accuracy in solution NMR as well [28,29].) Importantly, while NOE-derived interproton distance restraints scale with 1/r^6^ and have a limit of ~5.5 Å, dephasing in REDOR scales with 1/r^3^, allowing the determination of longer distances, which depending on the gyromagnetic ratio of the recoupled spin pair can reach 15–20 Å (Figure 2).

The accuracy of the interatomic distance obtained from REDOR depends on several factors (e.g., resonance offset effects, dipolar couplings other than the desired recoupled interaction, sample and instrument stability, B_1_ inhomogeneity, appropriate signal-to-noise, and proper natural abundance correction). Among these, soon after the development of REDOR, it was realized that the CSA of the dephasing spin can interfere with the REDOR measurement [30]. To eliminate the detrimental effect of the resonance offset of the dephasing pulses, several phase cycling schemes (xy-4, xy-8, and xy-16) were developed [26]. Among them, according to numerical simulations of, for instance, the effect of ^15^N CSA in frequency-selective (FS) ^13^C–^15^N REDOR experiments, the xy-4 scheme has been found to be the most adequate [31]. In the case of dephasing nuclei with a large CSA tensor such as ^19^F, removing the effect of CSA is more challenging, in particular at high magnetic fields, requiring composite pulses and/or high MAS rate [23,32].

As the determination of longer distances requires longer evolution times, achieving the longest possible T_2_ of the observed nuclei is of high importance. For this, high MAS and sufficient proton decoupling is necessary. While the *S_o_* reference experiments compensate for low proton decoupling during the delays between the rf pulses, insufficient decoupling during the dephasing pulses (which at longer evolution times becomes accumulated) may introduce distortions into the observed dephasing [31] highlighting the importance of powerful decoupling schemes. TPPM (two-pulse phase modulation) decoupling [33] of ~80 kHz, for instance, has been shown to be sufficient to minimize ^15^N–^1^H interactions in FS ^13^C–^15^N REDOR experiments [31]. Additional decoupling schemes and their combinations have been worked out over the years and used successfully for the elimination of the detrimental effect of the dense network of protons in, for instance, Y-detected ^1^H–X REDOR measurements [34,35,36] and in ^13^C-detected FS REDOR measurements aimed at the determination of ^1^H–^13^C distances [37]. Of note, as high-power proton decoupling may lead to sample heating, care has to be taken to monitor the sample integrity and probe detuning during the course of the experiment. Cycling between the *S* (dephased) and *S_o_* (reference) spectra upon collecting the data at each dipolar evolution time point is recommended, which takes care of issues such as instabilities in CP efficiency. More detailed considerations for sample preparation and experimental setup have been given in an excellent summary by Thompson and coworkers [38].

In addition to the determination of intramolecular and/or intermolecular distance restraints, REDOR is frequently used as a spectral editing tool. This is illustrated in Figure 3 for a site specifically labeled peptide embedded in lipid bilayers, where the ^13^C spectrum (S_o_) has contributions from the ^13^C label at Ala_10_ (residue *i*) as well as from natural abundance peptide and lipid carbonyl carbons [39]. The selection and unambiguous assignment of the labeled Ala_10_ carbonyl carbon resonance utilizes the strong one-bond dipolar coupling with [^15^N]Gly_11_ (residue *i + 1*) by a ^13^C{^15^N} REDOR or related transferred-echo double resonance (TEDOR) [40] experiment performed at short dipolar evolution time (<1 ms), where only ^13^C–^15^N atom pairs in close proximity to each other report. Accordingly, the resonance line in the difference spectrum (∆S) at 177 ppm arises exclusively from the carbonyl-carbon of Ala_10_. Inspired by the 1D applications, modified versions of REDOR have been developed and used as a building block in two-dimensional experiments to reduce the spectral overlap and facilitate the assignment of ^13^C, ^15^N-labeled proteins. The examples include the selection of methine signals of valine, leucine, and isoleucine residues by suppressing the overlapping methylene signals of longer side chains [41]. The full suppression of undesired signals is achieved by an asymmetric REDOR sequence, where every other 180° pulse is shifted (~0.2 τ_r_) from the middle of the rotor period resulting in a deeper minimum of the dephasing curve. A similar approach is used for the edition of side chain carboxyl signals of glutamate and aspartate residues by suppressing the overlapping backbone carbonyl peaks [41]. As illustrated by the applications in the following sections, spectral editing by one- and two-dimensional REDOR-based experiments enables (i) the determination of the chemical shift of specific nuclei, (ii) the selection of a specific nucleus for subsequent distance measurement by yet another (REDOR or else) recoupling scheme, (iii) the suppression of undesired signals and the reduction in spectral overlap, (iv) the quantification of specific functional groups in composition analysis, and (v) the characterization of motional processes in the protein backbone. Creative labeling strategies [42] are used for each application.

## 3. Variations on REDOR and REDOR-Based Multidimensional Solid-State NMR Measurements

Modifications of REDOR are motivated by two main desires: (i) the measurement of multiple internuclear distances simultaneously and (ii) increasing the limit of the measurable distances [44,45]. The desire to determine multiple internuclear distances using a single sample, has led to the development of frequency-selective (FS) REDOR pulse sequences amenable to uniformly ^13^C, ^15^N-labeled systems. First, a REDOR pulse sequence was combined with a frequency-selective spin echo to recouple a single ^13^C–^15^N dipolar interaction with a concomitant suppression of the remaining ^13^C–^15^N dipolar and all ^13^C-^13^C scalar couplings to the selected ^13^C spin [31]. The robustness of the measurement was demonstrated by the determination of distance restraints for the active site of light-adapted [U–^13^C, ^15^N] bacteriorhodopsin in its native purple membrane [46]. Another strategy to reintroduce the dipolar interaction between the observed nuclei and certain chemical moieties within a narrow frequency range of the nonobserved nuclei employs a so-called DANTE (“delays alternating with nutation for tailored excitation”) inversion [47,48]. This is exemplified by a study of the mechanism of inhibition of 3-deoxy-D-manno-2-octulosonate-8-phosphate (KDO8P) synthase, an enzyme involved in lipid A maturation and cellular growth in Gram-negative bacteria, where frequency selectivity was achieved by applying two rotor-synchronized DANTE *π*/2 pulses to the nonobserved nuclei, interleaved with an interval of one rotor period [49]. In the absence of DANTE recoupling pulses, the pulse scheme yields a full reference spectrum (*S_o_*), whereas its difference spectrum (Δ*S*) exhibits peaks of *S* nuclei due to dipolar coupling with spin I within a narrow chemical shift range. Inspired by these methodological achievements, several extensions of FS-REDOR have been developed including double single-quantum (DSQ)-REDOR directed at the measurement of weak intermolecular ^13^C–^15^N dipolar couplings in the presence of strong intraresidue ^13^C–^15^N couplings [50]. More recently, for probing a unique amino contribution in uniformly ^13^C, ^15^N-labeled *S. aureus* whole cells, a three-part relayed experiment including a FS-REDOR followed by a ^13^C mixing period by dipolar-assisted rotational resonance (DARR [51]) and a second FS-REDOR period has been implemented [52].

REDOR and its variations have also been incorporated as a building block into ^13^C–^13^C and ^13^C–^15^N correlation experiments aimed at the determination of backbone and side chain torsional angles in proteins. Most of these experiments correlate the relative orientations of two dipolar coupling tensors across the intervening bond. For example, the torsional angle ϕ can be determined from the relative orientations of the N–HN and Cα–Hα dipolar tensors extracted from ^15^N–^13^C multiple-quantum dipolar sideband spectra [53]. Briefly, the N–C dipolar interaction is recoupled by REDOR-type rotor-synchronized ^15^N and ^13^C 180° pulses followed by an evolution of the excited N–C double- and zero-quantum coherences under the C–H and N–H dipolar interactions. Subsequently, the N–C multiple quantum coherences are reconverted to single-quantum ^13^C coherence yielding sideband patterns separated according to the ^13^Cα isotropic chemical shift. A similar strategy is employed for the determination of the torsional angle ψ in ^13^C–^13^C correlation experiments, where REDOR-type 180° pulses sandwiched between two blocks of a C7 [54] or other homonuclear recoupling *rf* sequence are used for preventing the coherent averaging of the ^15^N–^13^C dipolar interaction [55]. A series of measurements with varying REDOR mixing times is carried out collecting an *S_o_* (without ^15^N dephasing pulses) and an *S* (with ^15^N dephasing pulses) spectrum at each dipolar evolution time yielding a ψ-dependent *S*/*S_o_* dephasing curve. Since their introduction, several versions of the HNCH and NCCN experiments described above have been developed with improved spectral resolution and targeting different ranges of the Ramachandran plot [56,57,58].

To push the limit of the measurable interatomic distance range to higher distances and facilitate the determination of backbone and side chain dihedral angles in proteins, REDOR methodologies have been extended to measure ^1^H–X dipolar couplings as well. In the experiment developed by Schmidt-Rohr and Hong [34], the ^1^H chemical shift is refocused by a single 180° pulse in the middle of the evolution period, whereas two 180° pulses per rotor period are applied to spin X. With a homonuclear decoupling scheme suppressing proton–proton dipolar couplings, the magnetization of each ^1^H spin is modulated by its coupling to spin X. The encoded ^1^H–X dipolar coupling of a specific proton is read out by transferring its magnetization to a directly bonded spin Y. As shown by a ^15^N-detected H^N^–^13^C(O) REDOR study of an elastin-mimetic peptide, H^N^–C(O) distances up to 6 Å are determined with high accuracy [59]. While the H^N^–^13^C(O) measurement is used to constrain the backbone dihedral angle ϕ, the side chain torsional angle χ_1_ can be extracted from a ^13^C_β_-detected H_β_–^15^N distance measurement [35]. While these experiments require suitably labeled protein, frequency-selective ^1^H–X REDOR measurements (with X as the observed nucleus) allow the observation of multiple side chain to backbone distance contributions simultaneously, providing a robust approach for the elucidation of dihedral angles and hydrogen bonds in systems with uniform labeling [37]. More recently, to improve spectral resolution while pushing the limit of measurable distances, FS-REDOR under high MAS was combined with ^1^H detection on a U–^13^C, ^2^H, ^15^N-labeled protein back exchanged with 30% H_2_O [60]. At the beginning of the experiment, ^1^H magnetization is transferred to ^15^N for indirect evolution. Following its transfer back to ^1^H, a REDOR period is added before detection to dephase the ^1^H magnetization with proximate ^13^C atoms. Peak intensities in the resulting ^1^H–^15^N correlation spectrum depend on the dephasing by the ^13^C nuclei. As demonstrated by the authors, band selectivity is achieved with frequency-selective soft 180° pulses applied to the ^13^C spins. In addition to C_α_- and C(O)-selective versions for monitoring backbone correlations, selective methyl dephasing for the elucidation of backbone and specific side chain dihedral angles, as well as selective aromatic dephasing for unveiling tertiary contacts, has been implemented.

In addition to ^1^H, another high-γ nucleus, which when introduced as a coupling partner increases drastically the measurable internuclear distance in recoupling experiments, is ^19^F. Other advantages of fluorine include its high sensitivity to the chemical environment and that it causes a small perturbation of the native structure. Additionally, it has no natural abundance background. REDOR experiments by reintroducing dipolar coupling between ^19^F–X spin pairs have been used to extract distance information up to ~15 Å (depending on the coupling partner) in biological systems and has particularly been useful for obtaining the distance and orientation restraints for membrane peptides/proteins and protein–ligand complexes. To overcome the detrimental effect of the large CSA of ^19^F on REDOR dephasing, Hong and coworkers have shown the effectiveness of a composite pulse and demonstrated how ^1^H–^19^F REDOR measurements can be used for determining the protein side chain conformation of bulky aromatic residues and restraining tertiary structure in specifically labeled systems [36]. In more recent studies, simultaneous measurements of many long-range distances have become available by using multidimensional methodologies suitable for peptides/proteins containing multiple fluorines. In one such example, ^13^C–^19^F REDOR is combined with the 2D ^13^C–^13^C correlation to resolve multiple distances (up to ~10 Å) in ^13^C-labeled proteins containing a small number of fluorines [32]. Its ^1^H-detected counterpart [23] (Figure 1B) provides even longer range distances (up to ~15 Å) in a 2D-resolved fashion with high accuracy. As shown by the authors, due to the long distance range and the abundance of protons, a single fluorine label in a well-chosen aromatic residue could yield more than 30 fluorine–amide NH distance restraints in perdeuterated and back-exchanged proteins. The ^1^H–^19^F distances are extracted from mixing-time-dependent S/S_o_ signal intensities and, together with ^13^C–^19^F restraints and chemical shift information, could successfully be used in the structure calculation of proteins. Recently, an improved, three-dimensional version [61] of the experiment has been developed with the capability of resolving and assigning distance restraints in multiply fluorinated proteins, saving sample and instrument time.

Importantly, variations on REDOR have also been used as a building block in ^13^C–^15^N or ^13^C–^13^C correlation experiments for the study of protein–protein interfaces. In the experiment called dubbed REDOR-PAINCP (“proton assisted insensitive nuclei cross-polarization”) [62], a U–^13^C, ^15^N-enriched protein (Figure 4A) (protein *A*, pink) is complexed with its U–^15^N-enriched binding partner (protein *B*, blue). A ^15^N–^13^C REDOR filter is applied to dephase the ^15^N magnetization of protein *A* followed by a transfer of the remaining ^15^N magnetization from protein *B* to the ^13^C atoms of protein *A* across the intermolecular interface. As demonstrated by the authors on two differently labeled fragments of thioredoxin, depending on the implementation of different filters, the dephasing profiles give rise to different kinds of cross-peaks in the resulting 2D REDOR-filtered spectra. For biomolecular systems involving a binding partner where isotopic labeling is currently not possible (e.g., microtubules), an alternative methodology involving a double-REDOR filter has more recently been developed [25] (Figure 1C). In that case, at the beginning of the experiment, a simultaneous ^1^H–^13^C and ^1^H–^15^N REDOR filter is applied to the complex of a U–^13^C, ^15^N-labeled protein (Figure 4B) (protein *A*, grey) with its unlabeled binding partner (protein *B*, white) dephasing the magnetization of all the protons in the labeled protein. The remaining ^1^H magnetization of the complex arising from protein *B* is transferred across the intermolecular interface back to protein *A* by ^1^H–^13^C or ^1^H–^15^N cross-polarization mapping the intermolecular interface. The residues located at the interface can be assigned by a subsequent homonuclear or heteronuclear correlation step.

Finally, we add that as dipolar coupling between directly bonded nuclei yields information on the amplitude of motion in the solid state, REDOR can also be exploited to investigate dynamic processes in proteins. Specifically, motion on a time scale shorter than the reciprocal of the dipolar coupling averages the coupling and affects the dephasing curve [63]. This has motivated the development of experimental protocols and computational methodology to extract order parameters from one-bond ^1^H-X heteronuclear couplings. To overcome the problem posed by fast dephasing as a result of strong one-bond ^1^H–X couplings, a modification of the original REDOR sequence has been introduced by shifting the position of one [64] or both [65] 180° pulses in the REDOR scheme. The combination of ε-REDOR [65] and DEDOR (deferred rotational-echo double resonance) [66] allowing the simultaneous measurement of ^15^N–^1^H and ^13^C_α_–^1^H_α_ dipolar couplings has been illustrated recently to provide residue-specific order parameters for the protein backbone for nondeuterated proteins at high MAS (100 kHz) [67]. While these ^1^H–X measurements are suitable for the characterization of motion from ps to tens of microseconds, one-bond ^13^C–^15^N dipolar couplings (~1 kHz) in uniformly ^13^C, ^15^N-labeled proteins can be exploited to investigate slower, ms time scale motions providing insight into conformational transitions. In the 2D REDOR-filtered DARR sequence implemented by Thompson and coworkers [68], REDOR filtering of C–C correlation spectra removes signals from rigid backbone carbons, whereas signals from backbone carbons with ms time scale dynamics become highlighted. Another recently developed approach for the identification of dynamic protein segments uses ^13^C–^2^H REDOR under slow to moderate MAS conditions by off-resonance irradiation [69]. The experiment is carried out in two parts. In the first part, ^2^H 180° pulses are applied far off resonance to invert deuterons with large quadrupolar couplings (i.e., with rigid ^2^H–^13^C bonds) and selectively recouple them to nearby carbons. In the second part, a near-resonance spectrum is collected where both rigid and dynamic moieties are recoupled. A difference of the two spectra highlights the signals of only mobile residues. As demonstrated on the β1 immunoglobulin binding domain of protein G (GB1), the experiment can be carried out in a two-dimensional ^13^C–^13^C correlation fashion to provide site-resolved dynamic information.

In the following sections, we survey the applications of REDOR and its novel implementations in the study of protein structure and function. Special attention will be given to isotopic labeling strategies. To illustrate the versatility of REDOR and the progress through the past few decades, we attempt to provide a historical perspective from the earliest to the most recent applications of complex biological systems.

## 4. Membrane-Associated Polypeptide Chains

Despite methodological advances in sample preparation and in experimental techniques [70,71,72,73], the elucidation of the structure and interaction of membrane proteins under near-physiological conditions remains a challenge. Solid-state NMR provides an opportunity to obtain high-resolution structural information for polypeptide chains embedded in membrane environments. After a pioneering work by Schaefer and coworkers on magainin 2, an antimicrobial peptide from the African clawed frog, *Xenopus laevis* [74], dipolar recoupling measurements by REDOR and related methodologies have been applied extensively to obtain site-specific information on the conformation and interactions of membrane peptides and transmembrane segments of larger proteins in lipid bilayers providing atomic-level insight into their mode of action. Furthermore, by the incorporation of REDOR as a building block into multidimensional ^13^C–^13^C and ^13^C–^15^N correlation experiments, intramolecular and intermolecular distance restraints are used for the complete structure determination of molecular assemblies in lipid environments.

### 4.1. Determination of Secondary Structure

Similarly to the observations in solution NMR, the chemical shifts of carbonyl and methyl carbons in the solid state are diagnostic markers of whether an amino acid residue is in an α-helical or β-sheet conformation [75]. Accordingly, chemical shift measurements are widely used to obtain information on the secondary structure and conformational changes in transmembrane proteins. As labeled ^13^C sites are often not shift-resolved from the large natural abundance lipid background, their unambiguous assignment requires a spectral filtering. As discussed above, this can be achieved by exploiting the ~1 kHz dipolar coupling between a labeled ^13^C(O)*_i_*–^15^N*_i+1_*atom pair selecting unambiguously the backbone carbonyl carbon resonance of residue *i* (Figure 3). In addition to the investigation of pore-forming antimicrobial peptides [39,43,76,77,78], such one-dimensional REDOR-filtering measurements have extensively been applied to viral fusion proteins to monitor secondary structure at different lipid compositions and lipid-to-peptide molar ratios [79] and, by focusing on specific sequential residue pairs, to identify conformational changes in bacteriorhodopsin [80]. An alternative strategy for the determination of peptide conformation is based on distance measurements between selectively labeled residues (Figure 5A,B). A good example is the investigation of phospholamban (PLB), the 52-amino-acid membrane-spanning protein with a primary role of regulating the active transport of calcium ions into the sarcoplasmic reticulum lumen via an inhibitory association with the Ca^2+^–ATPase SERCA. In a joint rotational resonance [14] and REDOR NMR investigation of PLB [81], site-specific ^13^C and ^15^N labels were placed in the peptide backbone, and the sample was reconstituted in multilamellar phospholipid vesicles in the presence and absence of the Ca^2+^–ATPase. Distance measurements between [2-^13^C]Ala_24_ and [^15^N]Gln_26_ revealed that SERCA induces a switch from an alpha-helical to a more extended conformation, suggesting that structural distortions in the juxtamembrane region may change the relative orientation of the TM and cytoplasmic domains of PLB promoting its association with the Ca^2+^–ATPase. A similar approach yet one that allows the monitoring of longer interatomic distances has been applied to EqtII_1-32_ [82], a peptide corresponding to the N-terminal region of equinatoxin, a member of the family of actinoporins. Specifically, ^19^F{^13^C} and ^19^F{^15^N} REDOR performed on [4-^19^F]Phe_16_-[1-^13^C]Leu_19_ and [4-^19^F]Phe_16_-[^15^N]Leu_23_ analogues of EqtII_1-32_ provided evidence of a shift toward a conformation with a longer helical stretch in sphingomyelin (SE)-containing phosphatidylcholine bilayers with possible implications for pore formation of EqtII in SE-containing membranes.

### 4.2. Location in the Lipid Bilayer

The phosphorous atom in phospholipid headgroups (a spin-1/2 isotope of natural abundance) is a natural label that can be utilized in mapping the location of nonphosphorylated polypeptide chains in lipid bilayers. By incorporating ^13^C or ^15^N labels at various locations in the peptide sequence, ^13^C{^31^P} and/or ^15^N{^31^P} REDOR experiments can be used to probe the proximity of specific sites to the lipid headgroups. We should note that since there are multiple phosphorous dephasers whose contributions are difficult to deconvolute, these experiments provide only a qualitative measure of the proximity to the membrane surface. Nevertheless, by probing the ^13^C–^31^P proximity at different sites along the peptide sequence [84], in particular, at both ends of a molecule, or monitoring the peptide–headgroup contact as a function of the peptide concentration [43], ^13^C{^31^P} REDOR can provide a valuable piece of information in differentiating between different pore models of cell-permeabilizing peptides (Figure 6). A similar experimental strategy has been used in establishing a correlation between the membrane insertion depth and the fusogenicity in various forms (wild type vs. mutated, monomer vs. trimer) of the N-terminal 25-residue segment of HIV glycoprotein 41 (gp41) [85] and to probe early stage interactions between the β-amyloid (Aβ) peptide and phospholipid headgroups in synaptic plasma membranes extracted from rat’s brain tissues exploring membrane-associated nucleation processes in fibrillation [86].

In addition to probing the proximity between the phospholipid headgroups and the peptide backbone, incorporation of site-specific ^13^C or ^15^N labels into the amino acid side chains can provide further information on peptide–lipid interaction. This was shown elegantly in the investigation of an arginine-rich cationic antimicrobial peptide, protegrin-1 (PG-1) [87], where by taking into account the effect of multiple ^31^P spins, it has been shown that guanidium groups can form a bidentate complex with two lipid headgroups stabilized by N–H…O–P hydrogen bonds and electrostatic interactions. Considering the transmembrane orientation of the PG-1 hairpin known from independent measurements [88], the short ^13^C_ξ_(Arg)–^31^P distances observed for multiple residues (4.0–6.5 Å) in anionic membranes have provided evidence for the formation of toroidal pores where some of the lipid molecules reorient themselves and together with the peptide chains form the wall of the pore (Figure 6C).

In addition to the phosphate headgroups, lipids fluorinated at a single site of the acyl chain [89,90] provide an additional structural probe for the insertion depth and lipid interactions of membrane-active molecules. As shown by Blazyk and coworkers, to avoid interdigitation, the mole fraction of fluorinated lipids should not exceed ~0.1 [91]. Another alternative is the utilization of deuterated lipids. The advantages include the chemical equivalence of ^1^H and ^2^D and the commercial availability of fatty acids with deuterium labels at specific positions along the acyl chain allowing their use as a spectroscopic ruler along the membrane normal. The latter was demonstrated by Weliky and coworkers [92] for a model peptide KALP with a known transmembrane orientation and used subsequently for the elucidation of the membrane location of the fusion peptide region of HIV gp41 [93].

The effect of the lipid composition and in particular the effect of cholesterol on pore formation and membrane fusion has long been a focus of interest [94]. In a recent application, ^19^F labels were introduced into the side chains of two aromatic residues ([4-^19^F]F673 and [5-^19^F]W680) in a 44-residue peptide corresponding to the membrane-proximal external region and the transmembrane domain (MPER-TMD) of gp41, and the trimeric fusion peptide was reconstituted in virus-mimetic phospholipid membranes containing ^13^C-labeled cholesterol [95]. To characterize gp41–cholesterol proximities, after the assignment of the fifteen ^13^C-labeled sites in cholesterol, ^13^C–^19^F REDOR experiments sensitive to distances up to ~10 Å were performed with both ^19^F detection of the peptide and ^13^C detection of the cholesterol. Analyses of the REDOR dephasing curves have suggested that three cholesterol molecules are bound near F673 in each gp41 trimer. Together with MD simulations, the REDOR measurements have indicated that the helix–loop–helix conformation of the trimeric MPER-TMD (with a surface lying MPER helix and transmembrane trimeric stalk) has the capability of sequestering cholesterol and suggested that by imposing a local membrane curvature and disorder, cholesterol binding to gp41 could have a direct role in mediating virus–cell fusion. The cholesterol binding site of the influenza M2 proton channel was investigated in lipid bilayers by similar measurements [96]. As shown by C–F REDOR between ^13^C-enriched M2 and chain-labeled F_7_–cholesterol, two cholesterol molecules bind each M2 tetramer. According to the REDOR measurements, cholesterol binds the C-terminal TM residues in the vicinity of an amphipathic helix without a specific cholesterol recognition motif. Together with orientational restraints inferred from deuterium NMR, the model of M2-bound cholesterol was established identifying the hydrophobic, aromatic, and polar interactions stabilizing the complex.

### 4.3. Association of Polypeptide Chains in Lipid Bilayers

In addition to elucidate the conformation of polypeptide chains, their location in the lipid bilayer, and interactions with membrane components, REDOR and other dipolar recoupling experiments have proven to be powerful approaches in the determination of intermolecular distance restraints between peptide chains situated in membrane bilayers (Figure 5C). These experiments have provided structural and mechanistic insights into the assembly and function of a wide range of biological systems including pore-forming antimicrobial peptides, ion channels, fusion proteins, and receptors.

In many applications, isotopic labels are incorporated into unique amino acid side chains and are chemical shift resolved, simplifying the analysis. This is exemplified by one of the earliest studies of the M2 proton channel [97] essential for influenza infection, where site-specific ^15^N and ^13^C labels were introduced at the side chains of His_37_ and Trp_41_, respectively, in the TM segment of the protein. An upper limit of 3.9 Å for the interhelical [^15^N_π_]His_37_-[^13^C_γ_]Trp_41_ distance, inferred from ^13^C{^15^N} REDOR measurements, in combination with the tilt and rotational angles of individual helices known from independent studies [98] has been used to identify side chain pairings and characterize side chain torsional angles for the tetrameric bundle. A similar approach was employed by Smith and coworkers [99] focusing on the juxtamembrane region of the phospholamban (PLB) pentamer. Three different samples were studied in which a specific glutamine residue (Gln_22_, Gln_26_, or Gln_29_) was specifically ^13^C or ^15^N labeled in its side chain, and the resulting PLBs were reconstituted in lipid bilayers in equimolar amounts. The strongest dipolar coupling was measured for Gln_29_, with an internuclear [5-^13^C]Gln_29_–[5-^15^N]Gln_29_ distance of 4.1 Å ± 0.2 Å followed by Gln_26_–Gln_26_ (~5.5 Å) and Gln_22_–Gln_22_ (>5.5 Å). The measured distances were consistent with an H-bonded network within the central pore of the pentamer and were used to establish the rotational orientation of the TM helices as they emerge from the membrane bilayer. A ^13^C/^15^N-labeling approach was applied also in a measurement series of the HIV-1 fusion peptide [100] providing experimental evidence of oligomer formation under fusogenic condition and showing the presence of both parallel and antiparallel arrangements of peptide chains. More recently, ^13^C{^15^N} REDOR measurements were used to confirm the conformation of a synthetic peptide with the capability of forming stable nanopores in lipid bilayers with practical applications such as single molecule detection and DNA sequencing [101].

While ^13^C–^15^N recoupling has a distance limit of ~5 Å, the incorporation of ^19^F labels and the employment of ^13^C–^19^F REDOR measurements substantially increase the limit of measurable intermolecular distances and widen the scope of answerable questions. This has been exploited in the study of a pore-forming antimicrobial peptide, (KIAGKIA)_3_ (K3) [39], illustrated in Figure 3. To unveil the pore structure and elucidate the mechanism of membrane permeabilization by K3, an equimolar mixture of peptide chains labeled by ^13^C or ^19^F at the Ala_10_ position were embedded in multilamellar lipid vesicles, and intermolecular ^13^C–^19^F distance measurements were performed. As discussed in the Introduction, a ^15^N label incorporated at the neighboring Gly_11_ was utilized for selection of the [1-^13^C]Ala_10_ signal before the ^19^F-dephasing step. The appearance of a sizeable ^13^C{^19^F} REDOR difference signal (Figure 3A, right) was a manifestation of the intermolecular [1-^13^C]Ala_10_–[3-^19^F]Ala_10_ dipolar coupling and proved the proximity of K3 chains in the lipid bilayer. The least-square analysis of the ^13^C{^19^F} REDOR curve with normalized Δ*S*/*S_o_* dephasing as a function of dipolar evolution time (Figure 3B) showed a bimodal distribution of [1-^13^C]Ala_10_–[3-^19^F]Ala_10_ distances: a narrow distribution centered at 4.6 Å suggesting specific interactions between the peptide chains and a broad distribution at longer distances arising from a less specific interhelical contact. Additional site-specific distance measurements together with differences in side band dephasing rates indicative of a preferred relative orientation between the CSA and the dipolar tensors [102] were used to obtain orientation restraints between the peptide chains yielding an approximate interhelical cross-angle of 20°. In conjunction with the ^13^C{^31^P} REDOR experiments and static NMR measurements probing the orientation of K3 chains with respect to the membrane normal as well as ^31^P{^19^F} REDOR showing increased lipid disorder in the presence of the peptide, a toroidal pore model was proposed, in which dimeric and monomeric K3 chains together with lipid headgroups constitute the pore (Figure 3C) [39,43].

In addition to ^19^F incorporation, another way to increase the distance limit in dipolar recoupling studies is the employment of ^1^H{^13^C} and ^15^N-detected ^1^H{^13^C} REDOR. These experiments are directed at the determination of the internuclear distance between a ^13^C label and an amide proton bonded to a ^15^N-labeled amide [34]. After performing a series of ^15^N{^13^C} and ^13^C{^19^F} REDOR experiments on mixtures of various combinations of site-specific ^15^N-, ^13^C-, and ^19^F-labeled protegrin-1 (PG-1), a β-hairpin AMP embedded in lipid bilayers, ^15^N-detected ^1^H{^13^C} REDOR measurements were performed on a 1:1 molar mixture of [^15^N]Cys_15_–PG-1 and [1-^13^C]Cys_15_–PG-1 to constrain the dimer interface and differentiate between various models [103]. The intermolecular distances inferred from REDOR indicated a parallel arrangement of the two protegrin-1 molecules with the C-terminal strand of the β-hairpin forming the dimer interface. The in-register arrangement of PG-1 chains has been shown to enhance the amphipathic character of the dimer and facilitate its insertion into the membrane.

Using similar strategies discussed above for peptides, REDOR and other homonuclear and heteronuclear dipolar recoupling techniques have successfully been used to investigate the mechanism of action of larger systems, such as membrane receptors and ion channels. In one such application, Thompson and coworkers [104] combined REDOR with site-directed mutagenesis to target the site of interest in the 60 kDa serine bacterial chemoreceptor by introducing a unique cysteine residue into one of the TM helices (*α*1) and the determination of interhelical distances between an engineered [1-^13^C]Cys_56_ and a fluorinated phenylalanine residue (Phe_163_) in α4. The ^13^C{^19^F} REDOR measurements have shown evidence of a change of 1.0 ± 0.3 Å in the distance between the two labeled positions upon ligation. Excluding the possibility of side chain motion, the authors have shown that the observed distance change arises from an approximate 1.6 Å intrasubunit piston motion of *α*4 upon serine binding. Other applications such as constraining the interface and rotational orientation of the TM segments of Neu receptor tyrosine kinase [105] and the investigation of the interactions of tryptophan residues in various states of the proton-motive photocycle of bacteriorhodopsin [106] have been reported.

In conjunction with other SS-NMR techniques, ^13^C{^19^F} REDOR has also contributed to elucidate the structure and mode of action of viral proteins. By introducing site-specific ^13^C and ^19^F labels at the boundary of the MPER and TMD in the trimeric HIV gp41 protein, the C–F distance measurements were used to obtain evidence of a turn between the MPER and the TMD helix with implications for virus–cell membrane fusion [107]. The other main focus of interest has been the mechanism of gating and activation of influenza M2 proton channels. By the investigation of the effect of pH on interhelical packing of the tetrameric bundle, Hong and coworkers have shown that unlike the alternating access mechanism of influenza A M2 [108], the TM helices of influenza B M2 undergo a symmetric scissor motion enlarging the pore diameter at low pH [109].

With the outbreak of the COVID-19 pandemic, proteins of the SARS-CoV-2 virus became potential drug targets and vaccine candidates. REDOR measurements have helped the determination of the oligomeric structure of the TM domain of SARS-CoV-2 envelope protein E (ETM), a homopentameric cation channel required for virus pathogenicity [110]. Briefly, an equimolar mixture of ^13^C-labeled ETM and 4-^19^F-Phe-labeled ETM embedded in lipid bilayers mimicking the endoplasmic reticulum Golgi intermediate compartment (ERGIC) membrane was used to derive interhelical ^13^C–^19^F distances between three fluorinated phenylalanines (Phe_20_, Phe_23_, and Phe_26_) and the proximate carbons of the neighboring chain (Figure 7). Together with additional distance and torsional angle restraints, 35 interhelical REDOR-derived ^13^C–^19^F distances were used in the structure calculation defining the five-helix bundle. According to the measurements, while the entry of the pore is lined by small residues, the central segment of the TM domain is dominated by bulky hydrophobic residues narrowing the pore diameter and dehydrating the pore. Intriguingly, the helices deviate from ideal, exhibiting a twist between residues Phe_20_ and Phe_23_, which may indicate that the two halves of the channel act semi-independently when interacting with other viral and host proteins [110]. In addition to establishing the structure of the pentamer, the authors have also used REDOR to investigate the interaction between ETM and hexamethlylene amiloride showing that the drug exchanges between multiple helices in a dynamic fashion and inhibits cation conduction by steric occlusion of the pore.

## 5. Amyloid Peptides

The conversion of proteins from their soluble states into fibrillar aggregates in the brain and other organs is associated with a wide range of pathological conditions including Alzheimer’s and Parkinson’s diseases, type II diabetes, and a number of systemic amyloidosis [111]. Solid-state NMR techniques have proven to be a powerful experimental approach for obtaining high-resolution structural information on the organization of insoluble fibrils produced by amyloid-forming peptides. Much of this work was pioneered in Robert Tycko’s laboratory. The idea is similar to what has been described for membrane peptides. By inserting site-specific isotopic labels at strategically chosen locations into amyloid-forming peptides, dipolar recoupling measurements can be used to gain information on the organization of the aggregates. This is demonstrated by a ^13^C{^15^N] REDOR study of a peptide representing residues 16–22 of the full-length β-amyloid (Aβ) peptide [83]. After confirming fibril formation by EM and X-ray diffraction, an antiparallel arrangement of beta-strands was elucidated by REDOR (Figure 5B). For this, a ^13^C and a ^15^N label were incorporated at the carbonyl carbon of Leu_17_ and the amide nitrogen of Ala_21_, respectively. While in a parallel beta-sheet arrangement, the shortest ^13^C–^15^N distance was expected to exceed 8 Å, corresponding to no measurable dephasing; in an antiparallel beta-sheet arrangement, the expected distance between the labeled sites is much shorter. The observed strong dephasing corresponding to a ^13^C–^15^N distance of 4.4 Å provided evidence of the antiparallel arrangement for Aβ(16–22). In a subsequent work [112], increasing the amphiphilicity of Aβ(16–22), the authors have found a change in the organization of the β-sheet from antiparallel to parallel, indicating that amphiphilicity plays a key role in the structural organization of amyloid fibrils. In a more recent study, REDOR and other solid-state NMR measurements were applied to microtubule-associated protein tau [113]. Specifically, the ^13^C–^15^N distance measurements in cross-β fibrils provided insight into how tau isoforms containing three (3R) or four (4R) microtubule-binding repeats are organized in Alzheimer’s tau tangles. The authors show that the two isoforms are mixed nearly equimolarly and exhibit an isoform-independent preservation of the β-sheet conformation, which, as they hypothesize, may have a role in the predominance of Alzheimer’s disease among neurodegenerative disorders.

Another recent study, along with neutron scattering and atomic force microscopy, employed ^13^C{^15^N} REDOR to elucidate the packing mode of an antiparallel β-sheet trimer of a synthetic amphiphatic peptide, Ac-I_3_VGK-NH_2_, forming flat nanoribbons [114]. The extensive nonpolar zipper between the β-strands unraveled by the distance measurements exhibited a two-residue shift leading to β-sheet lamination consistent with the observed architecture of the nanoribbon assembly. It is noteworthy that while many peptides form fibril structures that tend to maximize contacts among hydrophobic residues, peptides with low hydrophobicity employ alternate stabilizing interactions. This has been shown for the 20-residue peptide corresponding to an asparagine- and glutamine-rich segment of the yeast prion protein Ure2p [50]. DSQ-REDOR, a modification of REDOR and FS-REDOR allowing the measurement of weak intermolecular ^13^C–^15^N dipolar couplings in the presence of stronger intraresidue ^13^C-^15^N couplings, provided evidence for hydrogen bonding between glutamine side chains in adjacent parallel β-strands in Ure2p(10–39) consistent with the “polar zipper” model of stabilization by Perutz [115].

For the development of efficient imaging agents, understanding the structural basis of their interaction with the target protein is required. This is particularly challenging in the case of agents targeting large insoluble protein assemblies such as amyloid fibrils. In a recent study, Hong and coworkers [116] used ^13^C and ^19^F solid-state NMR to investigate the binding sites of a positron emission tomography ligand, flutemetanol, to the 40-residue Aβ peptide. In addition to the characterization of the binding stoichiometry and the flutemetanol-bound conformation of Aβ40 fibrils by solid-state NMR, they used a combination of various one- and two-dimensional (^13^C{^19^F}, ^13^C-detected ^1^H{^19^F}, and ^13^C–^13^C resolved ^1^H{^19^F}) REDOR measurements to identify and characterize the flutemetanol-interacting residues. REDOR-constrained docking simulations have indicated that the three identified closest lying segments (^12^VHH^14^, ^18^VFF^20^, and ^39^VV^40^) of Aβ40 constitute multiple binding sites. Based on the experimental data and simulation, the authors proposed a model of the bound flutemetanol.

## 6. Biomineralization Peptides

Biomineralization in hard tissue is controlled by extracellular matrix proteins, which exert their action by either triggering cell signaling pathways or directly promoting or inhibiting the growth of minerals. Solid-state NMR is well suited for the study of protein–surface interactions and can provide molecular-level insights into the recognition of biomineral surfaces. Statherin, a 43-amino acid salivary protein, has a dual function of inhibiting both the nucleation and the crystal growth of hydroxyapatite (HAP), Ca_10_(PO_4_)_6_(OH)_2_, the main mineral component of bone and teeth, via a direct interaction with the HAP surface. In a series of investigations by Drobny and coworkers, dipolar recoupling techniques in conjunction with NMR relaxation experiments were used to characterize the conformation and dynamics of HAP-bound statherin and its N-terminal 15-residue segment, SN15 [117,118]. In addition, to probe the side chain interactions of SN15 with HAP, intermolecular distances were measured from ^15^N and ^13^C labels in the side chains of basic [119,120], acidic [121], and aromatic [122] residues to ^31^P spins on the HAP surface. To explore how the size of the dephasing ^31^P spin network influences the REDOR curve, the authors performed ^31^P–^31^P recoupling experiments on HAP itself and ^15^N{^31^P} REDOR on (NH_4_)_2_HPO_4_, showing that the ^31^P–^31^P dipolar coupling between phosphorous nuclei in HAP can be replaced by an effective dipolar interaction [123]. Using similar isotopic labeling strategies and dipolar recoupling techniques, the structure, orientation, and dynamics of the leucine-rich amelogenin protein (LRAP), the predominant protein involved in enamel development, was investigated at the HAP interface under various conditions of pH and ionic strength [124,125]. Additionally, the effect of phosphorylation of a specific serine residue was explored on the LRAP–HAP interaction. Distance restraints inferred from ^13^C{^15^N} and ^13^C{^31^P} REDOR measurements probing the secondary structure and the LRAP–HAP interaction, respectively, have shown that ionic strength and dephosphorylation at pS16 may trigger a switching mechanism affecting the N-terminal segment (Leu_15_–Ser_28_) of LRAP during enamel development [125].

## 7. Protein–Ligand and Protein–Nucleic Acid Interactions

### 7.1. Protein–Ligand Distance Restraints

By the determination of specific protein–ligand distance restraints, REDOR has proven to be a powerful tool for the investigation of the active sites of enzymes. Some of these applications take advantage of the existence of unique, chemical shift resolved side chains in the binding pocket as was the situation in one of the earliest works by Schaefer and coworkers on the glutamine-binding protein (GlnBP) of *E. coli*, where the existence of a single histidine residue in the binding site was exploited in the derivation of distance restraints between [U-^15^N]GlnBP and L-[5-^13^C]Gln [126]. The investigation of the mechanism of action of 3-deoxy-D-*manno*-2-octulosonate-8-phosphate synthase (KDO8PS), the enzyme responsible for the biosynthetic formation of the unusual eight-carbon sugar 3-deoxy-D-*manno*-2-octulosonate required for lipid A maturation and cellular growth in Gram-negative bacteria, used a more extensive multinuclear approach [127], where the combination of ^15^N{^31^P}, ^15^N{^13^C}, and ^13^C{^31^P} REDOR experiments were applied to the binary complexes of KDO8PS with its natural substrates, phosphoenolpyruvate (PEP) and arabinose-5-phosphate (A5P). The strategy included the preparation of uniformly ^15^N-labeled and [η-^15^N_2_]Arg-labeled enzymes and monitoring the ^15^N enzyme nuclei as a function of their dipolar interactions with proximate ligand ^31^P and ^13^C nuclei. In parallel, ^31^P and ^13^C NMR experiments were conducted that directly monitored the state of the ligands themselves and their proximity to the enzyme ^15^N nuclei. The REDOR experiments established that PEP and A5P are bound by KDO8PS via two distinct sets of Arg and Lys residues corresponding to adjacent subsites in the enzyme. The solid-state NMR data complemented the crystallographic structure of KDO8PS and in combination with mutagenesis results allowed the identification of specific active site residues. Subsequently, frequency-selective REDOR was used to elucidate the action of a KDO8PS inhibitor mimicking the binding of A5P [49].

A similar multinuclear REDOR approach was used to characterize the ternary complex of 5-enolpyruvylshikimate-3-phosphate (EPSP) synthase with shikimate-3-phosphate (S3P) and N-(phosphonomethyl)-glycine (glyphosate or Glp) [128,129,130]. Glp binds to EPSP synthase in the presence of S3P and inhibits the condensation of S3P and phosphoenolpyruvate to form EPSP, a key intermediate in the synthesis of aromatic amino acids in plants. To obtain a high-resolution atomic-level picture of the binding site, the ligand–protein and inter-ligand distances were obtained from the combination of measurements employing specifically ^13^C and ^15^N-labeled Glp, biosynthetically ^13^C-labeled S3P, and a variety of ^15^N and ^19^F-labeled EPSP synthase analogues and taking advantage of native ^31^P in S3P and Glp. In another example by Schaefer and coworkers [131], the catalytic mechanism of lumazine synthase, an enzyme required for riboflavin synthesis in plants and microorganisms, was investigated by the analysis of a series of metabolically stable phosphonate reaction intermediate analogues complexed to the enzyme. The distance restraints from the phosphorous atoms of the ligands to the side chain nitrogens of specific lysine, arginine, and histidine residues determined from ^15^N{^31^P} REDOR experiments in combination with the X-ray crystal coordinates of one of the complexes were used in MD simulations to obtain an atomic-level picture of the enzymatic reaction.

In a more recent work by Hong and coworkers, the structure and dynamics of substrate-bound EmrE (“efflux-multidrug resistance E”), a member of the Small Multidrug Resistance transporter family in *E. coli*, was studied by solid-state NMR in lipid bilayers at two different pH values (Figure 8) [132,133]. After the chemical shift assignment, 2D ^1^H–^15^N-resolved ^1^H{^19^F} REDOR was employed to obtain distance restraints between EmrE and a fluorinated substrate, tetra(4-fluorophenyl) phosphonium (F_4_-TPP^+^). Together with torsional angle restraints, more than 200 H^N^–F protein–substrate distances (ranging between 5.8 and 12 Å) were used to solve the structure of the EmrE–TPP^+^ complex identifying key side chain–substrate interactions in the binding pocket and showing that at low pH, F_4_-TPP^+^ lies closer to Glu_14_, a key determinant of the proton-coupled substrate transport, in subunit A than in subunit B [132]. This was found to be in contrary to high pH, where the authors found evidence of a more symmetric binding site with the substrate being similarly exposed to both sides of the membrane [133]. The two ss-NMR studies have offered explanation for the observed asymmetric protonation of Glu_14_ [134] and provided high-resolution insight into the proton-coupled efflux mechanism of EmrE.

### 7.2. Conformation of Bound Ligands

In the examples discussed above, REDOR was used to identify residues involved in ligand binding and to infer specific intermolecular distance restraints between the enzyme and its substrate. Another key issue in elucidating the mechanism of enzyme reactions and of protein–ligand interactions in general is the determination of bound ligand conformations. In systems inaccessible to X-ray crystallography and solution NMR, REDOR measurements provide a powerful alternative.

One of the most extensively studied ligand conformation is that of the microtubule-bound anticancer drug paclitaxel (Taxol). While paclitaxel has a relatively rigid tetracyclic ring system, its four flexible side chains give rise to a variety of possible conformations posing a challenge for crystallographic studies. First, by introducing two site-specific ^13^C labels, a ^15^N label (adjacent to ^13^C), and a ^19^F label into paclitaxel, a double-REDOR approach was used to select the specifically labeled ^13^C sites of the drug in the megadalton natural abundance ^13^C background of microtubules followed by ^19^F dephasing [135]. The determined two ^13^C–^19^F distances between the side chain and the benzoyl moiety of paclitaxel were used as constraints in molecular modeling to refine the conformation of the microtubule-bound ligand, yielding a T-shaped conformation. In a subsequent study, a site-specific deuteron and a fluorine label were introduced into paclitaxel with the advantage of no natural abundance contribution [136]. Due to the small quantity of the labeled paclitaxel (0.1 μmol) bound to the 100 kD αβ-tubulin dimer, the measurements involved a heroic accumulation of million scans corresponding to an 18-day experimental time for both the S_o_ and the S spectra. A control experiment of similar length was carried out to demonstrate the spectrometer stability. The performed ^2^H{^19^F} REDOR measurements provided additional evidence of the bound paclitaxel conformation, and based on the REDOR model, a structurally constrained potent analogue was synthesized by Ojima and coworkers [137,138].

REDOR and double-REDOR experiments have also been used to determine the conformation of bound inhibitor in a number of enzyme complexes including thermolysin, a zinc endoproteinase from *Bacillus thermoproteolyticus* [139], and human Factor Xa, a 45 kDa enzyme acting at the early stages in the blood coagulation cascade [140,141]. Moreover, REDOR has been used successfully for the determination of the bound ligand conformation in membrane environments. This is exemplified by the study of noncovalent H+/K+-ATPase inhibitor analogues in gastric membranes by ^13^C{^19^F} REDOR, where distance and orientation restraints between two distant moieties of the bound inhibitor were used together with site-directed mutagenesis and photoaffinity labeling to build a binding model [142]. In a more recent study of Na,K-ATPase, REDOR was used to determine structurally diagnostic distances between carbon sites and the three phosphorous sites of bound ATP [143]. Specifically, exploiting the chemical shift resolved signal of C8 of [U-^13^C, ^15^N]ATP, frequency-selective DANTE-^31^P{^13^C}-REDOR was applied to determine the P_α_–C8, P_β_–C8, and P_γ_–C8 distances. The distributions of nine selected torsional angles consistent with the three determined distances in a set of 200,000 ATP conformations computed stochastically were narrowed by a systematic analysis of preferred conformations of adenyl nucleotide ligands available in the PDB database. Based on the constrained analysis of bound ATP conformations, a model of the nucleotide binding site of Na,K-ATPase was developed providing an atomic-level insight into molecular recognition between the membrane-embedded ATPase and its substrate.

More recently, REDOR in conjunction with MD simulations has shed light on the structural determinants of ligand recognition in protein kinase C (PKC), a family of signaling proteins associated with a number of cellular functions. In a study by Cegelski, Schaefer, and coworkers [144], site-specific ^13^C, ^19^F, and ^2^H labels were introduced into bryolog 1, a high-affinity analogue of the PKC modulator, bryostatin 1, and the labeled compound was complexed with the PKCδ-C1b domain in the presence of lipid bilayers. The ^13^C–^19^F, ^13^C–^2^H, and ^2^H–^19^F distance measurements between the labeled sites suggested a distribution of conformations for bound bryolog 1 (Figure 9), highlighting the dynamic nature of the system. Intriguingly, in addition to the major long-distance component, the fitting of the dephasing curves indicated a minor short-distance component consistent only with an intermolecular contribution to the REDOR dephasing. Additionally, the preferential dephasing of spinning side bands in the ^2^H{^19^F} REDOR spectra indicated an orientational preference for the ^2^H–^19^F internuclear vector. These findings suggest that some of the PKCδ-C1b-bryolog 1 complexes dimerize in the membrane by specific protein–protein interactions. The REDOR-derived distance restraints were used in MD simulations yielding a model of the membrane-bound PKCδ–C1b–ligand complex.

### 7.3. Protein–Nucleic Acid Interactions

Spectral overlap often poses a challenge for solution NMR studies of nucleic acid complexes. Dipolar recoupling techniques in the solid state provide an alternative for obtaining atomic-level information on drug–DNA, RNA–peptide, and enzyme–nucleotide complexes. Most of the published works so far use the approach of site-specific labeling and establishing distance restraints between specific functional groups of the binding partners. To avoid spectral overlap, shift-resolved ^31^P and/or ^19^F sites are preferred, such as in a study by Drobny and coworkers of the interaction of the HIV transactivation response element (TAR) RNA with the viral regulatory protein tat [145]. Specifically, ^31^P{^19^F} REDOR between phosphorothioate (pS)-incorporated 5′ of A27 and 2′-fluoro-2′-deoxyuridine introduced at the bulge residue U23 of TAR indicated a marked rearrangement (~4 Å P–F distance change) of the binding site upon the addition of a peptide comprising residues of the tat binding domain. In a subsequent study [146], a specific arginine residue of the peptide, Arg_52_, was uniformly ^13^C- and ^15^N-labeled, and 5-fluorouridine was incorporated at the U23 position of TAR. Intermolecular ^13^C–^19^F and ^15^N–^19^F distances inferred from REDOR measurements provided direct structural evidence for the proximity of Arg_52_ and U23. Additional distance restraints between Arg_52_ and the P22 and P23 phosphate groups in TAR were used to refine the model [147].

In another study, REDOR measurements by Schaefer and coworkers complemented solution NMR and binding affinity measurements to probe the mechanism of uracil recognition and the pathway of base flipping by uracil DNA glycosylase (UDG), an enzyme that selectively recognizes and excises unwanted uracil bases from DNA [148]. By introducing a difluorophenyl nucleotide, an analogue of uracil, the authors characterized the molecular details of how UDG achieves its specificity by the recognition of the shapes and certain types of molecular defects. The ^31^P{^19^F} REDOR data were used to narrow down which of several structural models for the enzyme-complexed DNA were appropriate by comparing the experimental REDOR curve with the calculated dephasing values based on the atomic coordinates of the structural models.

## 8. Cell Wall Architecture and Glycopeptide Antibiotics

With the emergence of glycopeptide-resistant enterococci and staphylococci phenotypes, elucidation of the mode of action of vancomycin, the last-resort glycopeptide antibiotics targeting the cell wall of Gram-positive bacteria, and the development of derivatives with improved potency have become of vital importance. After the first applications of REDOR for the investigation of cross-linking in the cell walls of *Bacillus subtilis* [149], Schaefer and coworkers have pioneered the development of methodology for the characterization of cell wall architecture and glycopeptide binding sites in intact cell walls and whole cells of *Staphylococcus aureus* [150,151]. The idea includes the incorporation of specific ^13^C- and ^15^N-labeled atom pairs diagnostic of the peptidoglycan (PG) lattice architecture into strategically chosen positions of the PG stem, cross-link, and glycyl bridge (Figure 10) by growing *S. aureus* in defined medium supplemented with isotopically labeled amino acids. For instance, the distance between D-[1-^13^C]Ala from the stem in one glycan chain to L-[^15^N]Ala in the stem of a neighboring glycan chain was determined by growing *S. aureus* in defined medium with D-[1-^13^C]Ala and L-[^15^N]Ala in the presence of alanine racemase inhibitor and performing ^13^C{^15^N} and ^15^N{^13^C} REDOR measurements. With the intramolecular ^13^C–^15^N distance >10 Å in the *L*-Ala-*D-iso*-Gln-*L*-Lys-*D*-Ala-*D*-Ala PG stem beyond the detection limit, the 4.4 Å ^13^C–^15^N distance reported by REDOR is assigned to intermolecular (between stems) *L*-Ala–*D*-Ala proximities [152]. Similarly, as at short (<1 ms) dipolar evolution times exclusively one-bond ^13^C–^15^N couplings are detected, a *D*-[1-^13^C]Ala and [^15^N]Gly pair of labels was used as a direct measure of PG cross-linking (the number of peptide stem D-alanines covalently linked to a pentaglycyl bridge) between stems [151]. Likewise, a [1-^13^C]Gly and L-[ε-^15^N]Lys amino acid pair was used for the quantification of bridge links (the number of peptide stem lysines covalently linked to a pentaglycyl bridge) at various stages of cell growth and upon treatment with antibiotics. Using such combinations of isotopic labels, the authors have shown that in *S. aureus,* vancomycin is likely to interrupt PG synthesis by interference with transglycosylation (glycan chain extension) rather than transpeptidation (reducing the number of cross-links between stems).

Subsequent REDOR experiments were used to measure site-specific internuclear distances between potent ^19^F-labeled glycopeptide antibiotics to ^13^C and ^15^N labels biosynthetically incorporated into the bacteria from labeled alanine, glycine, or lysine in the growth medium. One such example is a fluorobiphenyl derivative of chloroeremomycin, a potent derivative of vancomycin, where the combinations of ^13^C{^19^F}, ^15^N{^19^F}, and ^31^P{^19^F} REDOR measurements in isolated cell walls were consistent with positioning the vancomycin cleft of the antibiotic around an un-cross-linked D-Ala-D-Ala peptide stem with the fluorobiphenyl moiety near the base of a second, proximate stem of the PG matrix [150]. In a similar study of ^19^F-labeled oritavancin (Figure 10), a vancomycin analogue exhibiting antimicrobial activity against vancomycin-resistant Gram-positive strains, a ^15^N to ^13^C TEDOR recoupling exploited the one-bond ^13^C–^15^N dipolar coupling in the D-Ala-Gly cross-link in whole cells of *S. aureus* to select the D-[1-^13^C]Ala label of the PG stem from the natural abundance ^13^C background before ^19^F dephasing [154]. The ^13^C–^19^F distances between the labeled site of the PG stem and the ^19^F atom in oritavancin were used as constraints in MD simulations for docking the antibiotic in the cell wall of *S. aureus*. Subsequent studies have provided evidence that unlike vancomycin, oritavancin exhibits a dual mode of action against *S. aureus* by inhibiting both transglycosylation and transpeptidation [155], suggesting that in addition to the D-Ala-D-Ala pentapeptide stem terminus, it has a secondary cell wall binding site at the pentaglycyl bridging segment.

To explore the effect of PG chemical structure on cell wall architecture, the compositions of *S. aureus* and *E. faecium* cell walls, differing in their bridge structure, were characterized by solid-state NMR [156,157] using a similar methodology to that described above. As revealed by ^13^C{^15^N} and ^15^N{^13^C} REDOR measurements, the difference in the PG bridge length (pentaglycine in *S. aureus* vs. a single *D*-Asp in *E. faecium*) has a profound effect on bridge- and cross-link densities with implications for the frequency of stems terminating in glycopeptide-binding *D*-Ala-*D*-Ala units. By the analysis of vancomycin and oritavancin binding in *E. faecium*, the authors have also shown that while vancomycin acts by the inhibition of transglycosylation, oritavancin acts by the inhibition of transpeptidation [157]. To gain more insight into the binding interaction of oritavancin, whole cells of *E. faecium* enriched with L-[1-^13^C]Lys were complexed with desleucyl [^19^F]oritavancin, an analogue with a damaged *D*-Ala-*D*-Ala binding pocket. The ^13^C{^19^F} REDOR measurements have shown that the defected analogue still binds to PG, specifically, to nascent and template PG, supporting the hypothesis that the inhibition of transpeptidation in *E. faecium* is the result of a large number of secondary binding sites relative to the number of primary binding sites.

To further characterize the effect of the bridge length on PG architecture, isogenic Fem (factors essential for methicillin resistance)-deletion mutants of *S. aureus* have been investigated by solid-state NMR [158]. The advantage of using these constructs, as pointed out by the authors, was that they retain identical PG biosynthesis machineries, except for the *fem* deletions, which result in shortened bridge lengths. For example, instead of (Gly)_5_ in the wild type, FemB has (Gly)_3_, whereas FemA has (Gly)_1_ as its bridge structure. According to these measurements, the bridge length is a key factor in determining the level of cross-linking. Furthermore, the dramatic drop in cross-linking density observed in FemA (from 75% in WT to 50% in FemA) suggested fundamental differences in the PG structure indicating that the monoglycyl bridge in FemA is too short to form a parallel PG stem architecture existing in the wild type. Of note, solid-state NMR in conjunction with mass spectrometric analysis suggests some degree of heterogeneity in the PG of FemA as it consists of not exclusively monoglycyl bridges but rather both monoglycyl and triglycyl bridges [159]. In a subsequent in-depth structural study of the FemA mutant, the REDOR of isolated cell walls was used to measure the internuclear distances between ^13^C-labeled alanines and ^19^F-labeled lysines incorporated into PG [160]. The observed ^13^C–^13^C and ^13^C–^19^F distances were consistent with a tightly packed, hybrid architecture containing both parallel and perpendicular stems in a repeating structural motif of the PG lattice. For a more detailed summary of solid-state NMR approaches addressing the peptidoglycan architecture of Gram-positive bacteria, see the review by Kim and coworkers [152].

A more recent work provides insight into the applications of REDOR in the study of *S. aureus* biofilms [161]. Rather than a direct detection of surface proteins themselves (attached to the terminal glycyl residues of non-cross-linked *peptidoglycan* bridges), the authors used a strategy of their quantitation during the initial stages of biofilm formation by using a special isotopic labeling scheme. Briefly, [^15^N]Gly and l-[1-^13^C]Thr were incorporated to quantitate the fraction of peptidoglycan pentaglycyl bridge terminal -NH_2_ units covalently linked by sortase enzyme via Thr–Gly cell wall links to a surface protein. The increase in peptidoglycan stems that have bridges connected to a surface protein was determined by a cell wall *double* difference, where the planktonic cell ^13^C{^15^N} REDOR difference was deducted from the mature biofilm ^13^C{^15^N} REDOR difference. Additionally, a ^13^C leucine label was used to quantitate the increase in leucine-rich β-strand regions available for protein–protein and protein–surface binding in *S. aureus* biofilms promoting cell adhesion [161]. Solid-state NMR strategies for the composition analysis of bacterial biofilms have also been presented for *E. coli* [162] and *V. cholera* [163].

## 9. Concluding Remarks and Outlook

With advancements in instrumentation, methodology, and isotope labeling techniques, the past three decades have witnessed enormous progress in atomic-level structural and dynamic studies by ss-NMR. Together with chemical shift and orientation restraints, internuclear distances represent key structural information in deciphering the architecture of molecular assemblies and obtaining a mechanistic insight into protein interactions. REDOR is a robust and versatile spectroscopic tool with the capability of providing both intramolecular and intermolecular distances in multicomponent, heterogeneous systems such as lipid bilayers, fibrils, cell extracts, and even whole cells. By integrating it into multidimensional pulse sequences and the extension of measurements to ^1^H–X dipolar couplings, REDOR-based techniques now enable the simultaneous measurement of multiple distances, which, depending on the coupling partners, can reach ~2 nm. The above survey illustrates how, either by probing specific diagnostic proximities in site-specifically labeled proteins or by the measurement of multiple distances in uniformly labeled systems, atomic-scale information is inferred on membrane proteins, protein–ligand interactions, oligomeric assemblies to facilitate the elucidation of the mechanism of transmembrane signaling, enzyme catalysis, substrate transport, viral entry, amyloid formation, and drug binding. While sensitivity is still a challenge for systems involving large, megadalton complexes, ^1^H detection under fast MAS and sensitivity enhancement techniques such as dynamic nuclear polarization (DNP) [164] holds the promise for new in vitro and in vivo applications of REDOR and related ss-NMR techniques [165,166] for the study of large, multidomain assemblies and their interactions in near-physiological environments, improving our understanding of cellular mechanisms at the molecular level.

## Figures and Tables

**Figure 1 ijms-24-13637-f001:**
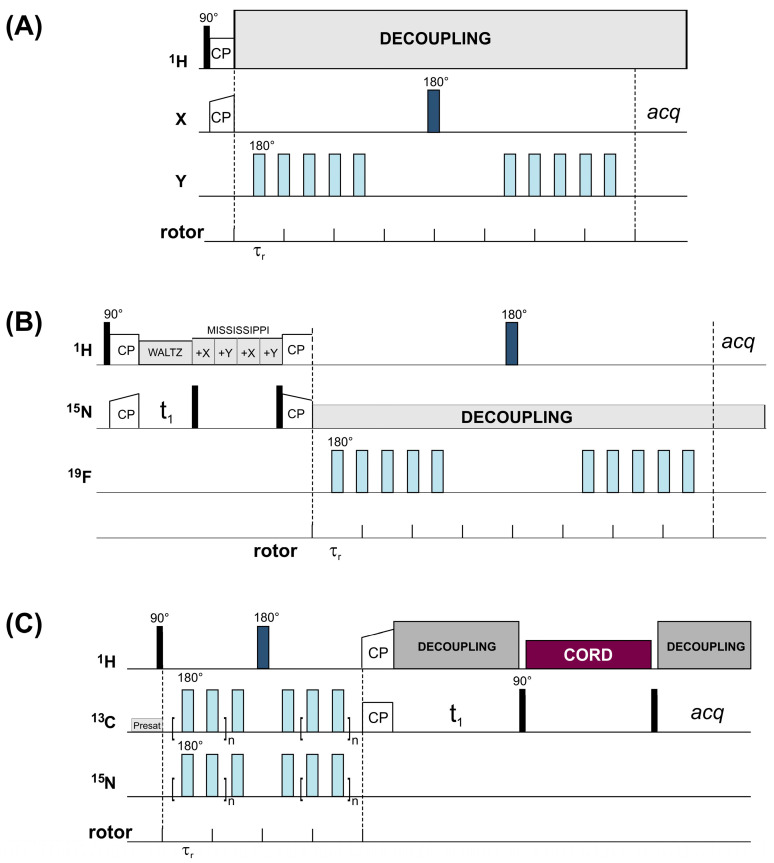
REDOR-based pulse sequences. (**A**) Basic 1D X{Y} REDOR for recoupling the X-Y heteronuclear dipolar interaction (where X and Y are typically ^13^C, ^15^N, ^31^P, or ^19^F) under MAS [16,17]. The experiment starts with cross-polarization (CP) transfer from protons establishing X magnetization. The application of 180° pulses every half rotor period on channel Y causes a net dephasing of the transverse X magnetization yielding a diminished intensity (*S* spectrum). In the absence of the 180° pulse train, a full echo is collected (*S_o_*). The difference spectrum (Δ*S* = *S_o_* − *S*) contains contributions only from those X nuclei that are dipolar coupled to Y. Dipolar coupling to protons is removed by high-power decoupling during the evolution and acquisition periods. (**B**) The 2D ^15^N–^1^H resolved ^1^H–^19^F REDOR for long-range ^1^H–^19^F distance measurement under fast MAS [23]. A REDOR block is inserted before the ^1^H detection, following a CP-based 2D heteronuclear single-quantum coherence (HSQC) sequence with a Multiple Intense Solvent Suppression Intended for Sensitive Spectroscopic Investigation of Protonated Proteins, Instantly (MISSISSIPPI) solvent suppression block [24]. The pulse sequence employs a ^1^H to ^15^N (beginning of the sequence) and a ^15^N to ^1^H (before the REDOR block) CP transfer step. As in (**A**), the experiment is carried out in two parts. The difference spectrum (Δ*S*) shows only the amide protons that are proximate to ^19^F. (**C**) A 2D ^13^C–^13^C correlation experiment with a double-REDOR block for the study of protein–protein interfaces [25]. Simultaneous ^1^H–^13^C and ^1^H–^15^N dipolar recoupling in the REDOR block employs an alternating xy-8 phase cycle [26] in the rotor-synchronized 180° pulse train. The ^13^C–^13^C homonuclear correlation is achieved by combined R*2_n_^v^-*driven (CORD) spin diffusion [27].

**Figure 2 ijms-24-13637-f002:**
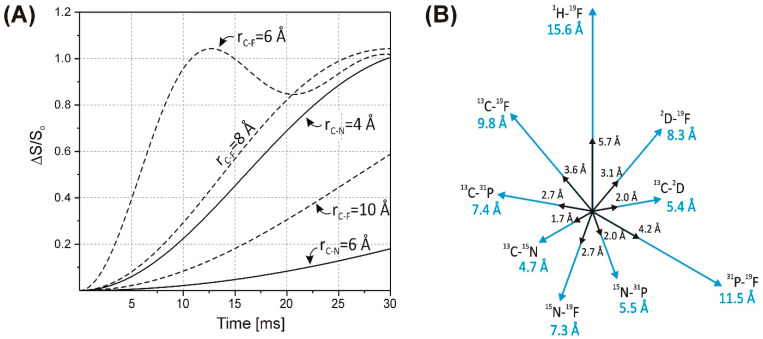
Distance measurements by REDOR. (**A**) REDOR dephasing curves for isolated ^13^C–^15^N (^____^) and ^13^C–^19^F (^......^) spin pairs at different distances. The initial slope depends on the gyromagnetic ratios of the coupled nuclei and the distance between them. For a given spin pair the slope is determined solely by the internuclear distance. (**B**) Representation of distances corresponding to dipolar couplings of 600 Hz (black) and 30 Hz (blue) between typical NMR-active nuclei used for the study of proteins.

**Figure 3 ijms-24-13637-f003:**
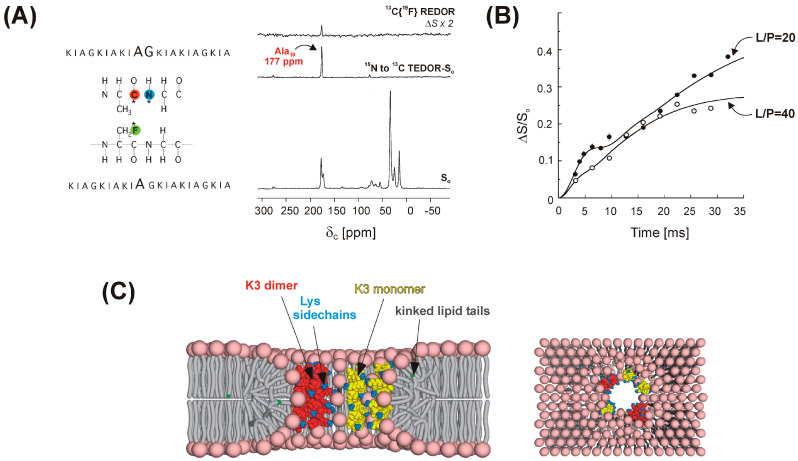
Dipolar recoupling for spectral editing and intermolecular distance restraints. (**A**) Positions of ^13^C, ^15^N, and ^19^F labels in two antimicrobial (KIAGKIA)_3_ (K3) peptide analogues used in REDOR experiments to probe the secondary structure and aggregation properties [39] (left). The ^15^N→^13^C{^19^F} TEDOR-REDOR NMR spectra of a 1:1 mixture of [1-^13^C]Ala_10_-[^15^N]Gly_11_-K3-NH_2_ and [3-^19^F]Ala_10_-K3-NH_2_ incorporated into multilamellar phospholipid vesicles (right). The full echo ^13^C spectrum (*S_o_*) shown at the bottom has contributions from the labeled ^13^C site and natural abundance peptide and lipid background. A short, four-rotor cycle (0.8 ms) ^15^N → ^13^C TEDOR coherence transfer selects the carbonyl carbon of Ala_10_ by exploiting the one-bond [1-^13^C]Ala_10_-[^15^N]Gly_11_ dipolar coupling (middle). Following the TEDOR selection, a 48-rotor cycle (9.6 ms) ^13^C{^19^F} REDOR is applied for probing the association of K3 chains. The Δ*S* spectrum arises exclusively from the dephasing of the TEDOR-selected [1-^13^C]Ala_10_ signal by ^19^F. Spectra were collected with magic angle spinning at 5000 Hz. (**B**) The ^13^C{^19^F} REDOR dephasing of a 1:1 mixture of peptides in (**A**) at lipid-to-peptide molar ratios of L/P = 20 and L/P = 40. The solid lines are simulated dephasing curves that correspond to a bimodal distribution of ^13^C–^19^F intermolecular distances (r_1_ = 4.6 ± 0.35 Å, r_2_ = 9.6 ± 1.5 Å at L/P = 20 and r_1_ = 4.3 ± 0.50 Å, r_2_ = 8.1 ± 1.6 Å at L/P = 40). At L/P  =  40, the population with the smaller mean and narrower distribution width decreases. (**C**) REDOR-derived model of the K3 membrane pore: cross-section *(*left*)* and top view *(*right*)*. K3 monomers and dimers are shown in yellow and red, respectively. Lysine side chains are indicated in blue. Peptide chains and lipid headgroups together line the wall of the torus-like pore. The ^13^C{^31^P} REDOR in conjunction with static NMR measurements in oriented lipid bilayers was used to elucidate peptide–lipid headgroup contact and orientation with respect to the membrane normal. Kinked lipid chains inferred from ^31^P{^19^F} REDOR measurements indicate increased membrane disorder in the presence of the peptide [39,43].

**Figure 4 ijms-24-13637-f004:**
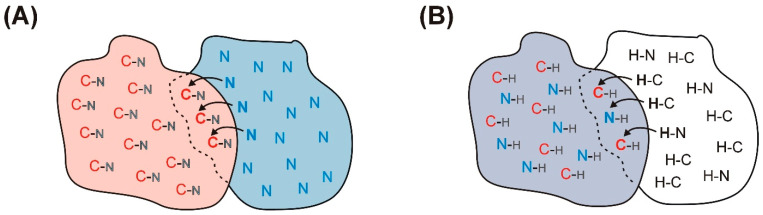
Cartoon representation of the application of REDOR as a filter in the elucidation of protein interaction surfaces. (**A**) A ^15^N–^13^C REDOR filter is applied to dephase the ^15^N magnetization of U–^13^C, ^15^N-enriched binding partner (protein *A*, pink) followed by a transfer of the remaining ^15^N magnetization from protein *B* (blue) to the ^13^C atoms of *A* across the intermolecular interface. (**B**) A simultaneous ^1^H–^13^C and ^1^H–^15^N REDOR filter is applied to dephase the magnetization of the protons of U–^13^C, ^15^N-enriched protein *A* (grey). Subsequently, the remaining ^1^H magnetization of the complex arising from protein *B* (white) is transferred across the intermolecular interface back to *A* by ^1^H–^13^C or ^1^H–^15^N cross-polarization highlighting the interaction surface.

**Figure 5 ijms-24-13637-f005:**
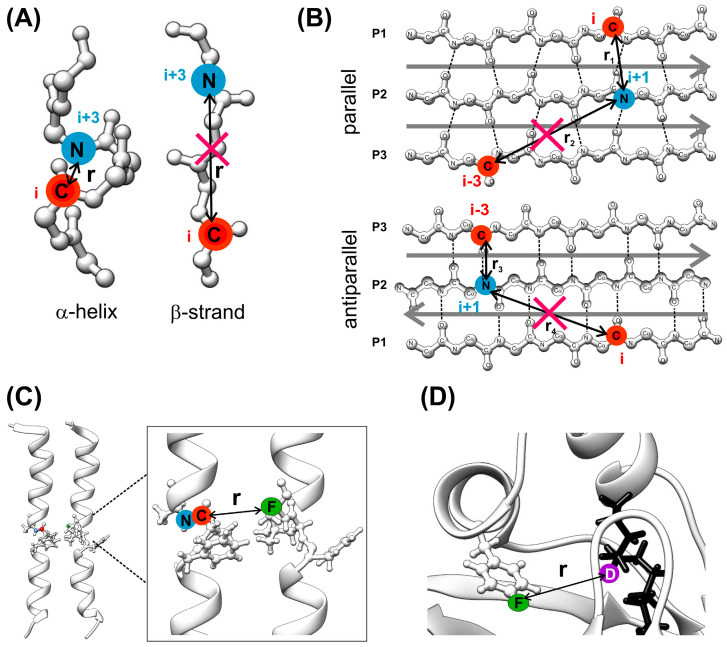
Probing the structure and interaction of polypeptide chains by REDOR. (**A**) A ^13^C and a ^15^N label introduced at residues *i* and *i + 3*, respectively, into the peptide backbone are used to distinguish between an α-helical or β-strand conformation. In an α-helix, the [1-^13^C]_i_–[^15^N]_i+3_ distance is expected to be ~3.7 Å, corresponding to a C–N dipolar coupling of 60 Hz giving rise to a sizeable dephasing. In a β-strand, the distance between the two labeled positions is ~7.9 Å (D_C–N_ = 6 Hz), which is beyond the limit of a measurable C–N distance; therefore, no dephasing is observed. (**B**) Distinguishing between a parallel or an antiparallel arrangement of β-strands using a combination of site-specific ^13^C and ^15^N labels. The ^13^C{^15^N} REDOR experiment on a sample containing an equimolar mixture of peptides P1 ([1-^13^C]Res(i)) and P2 ([^15^N]Res(i+1)) reports a measurable dephasing exclusively in a parallel arrangement. Sizeable dephasing between P3 ([1-^13^C]Res(i−3)) and P2 ([^15^N]Res(i+1)) indicates an antiparallel arrangement. A similar labeling scheme was used for the investigation of a seven-residue segment of the β-amyloid peptide by Tycko and coworkers [83]. (**C**) Probing the interhelical distance by REDOR. Preceeding a ^13^C{^19^F} REDOR measurement, the ^13^C label is selected from the natural abundance background by exploiting a one-bond ^13^C–^15^N dipolar coupling. Typically, a double-REDOR or a TEDOR-REDOR measurement is applied for selection and subsequent distance measurement. (**D**) Intermolecular protein–ligand distance measurement utilizing a site-specific ^19^F label incorporated into the side chain of a phenylalanine reside and a deuteron label in the ligand.

**Figure 6 ijms-24-13637-f006:**
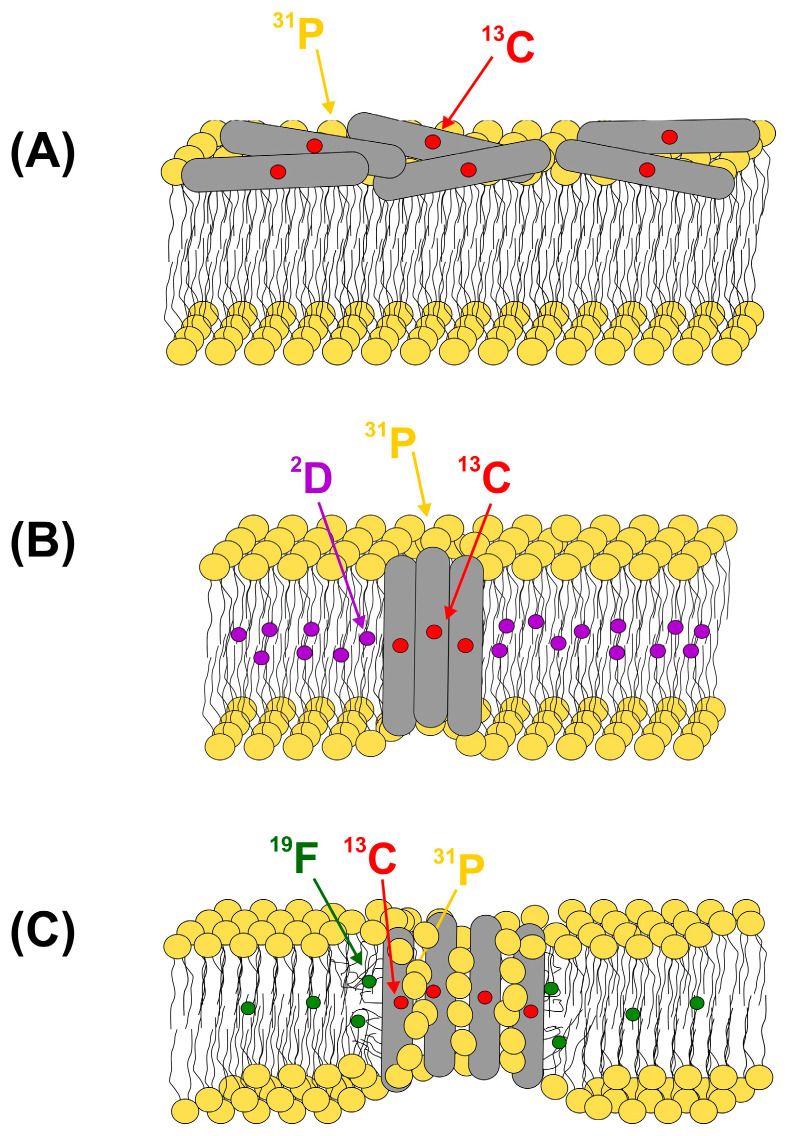
Distinguishing between location in the lipid bilayer and permeabilization mechanisms of membrane active peptides: (**A**) carpet, (**B**) barrel stave, and (**C**) toroidal mechanism. The ^13^C and/or ^15^N labels are placed at strategically chosen positions in the peptide chain. In addition to the natural ^31^P label of phospholipids, ^19^F and/or ^2^D labels can be inserted into the lipid acyl chains. Peptide–headgroup and peptide–lipid chain contacts inferred from ^13^C–^31^P, ^13^C–^19^F, and/or ^13^C–^2^D distance measurements are used as a ruler for the determination of the insertion depth of peptide chains in the membrane. Changes in the distribution of lipid headgroup–lipid tail distances assessed by ^31^P{^19^F} REDOR provide insight into peptide-induced membrane disorder. The association of peptide chains is probed by intermolecular distance measurements shown in Figure 3A,B. In addition to incorporating labels into the peptide backbone, ^15^N labeling of arginine and/or lysine side chains may provide further insight into the role of electrostatic interactions in pore formation.

**Figure 7 ijms-24-13637-f007:**
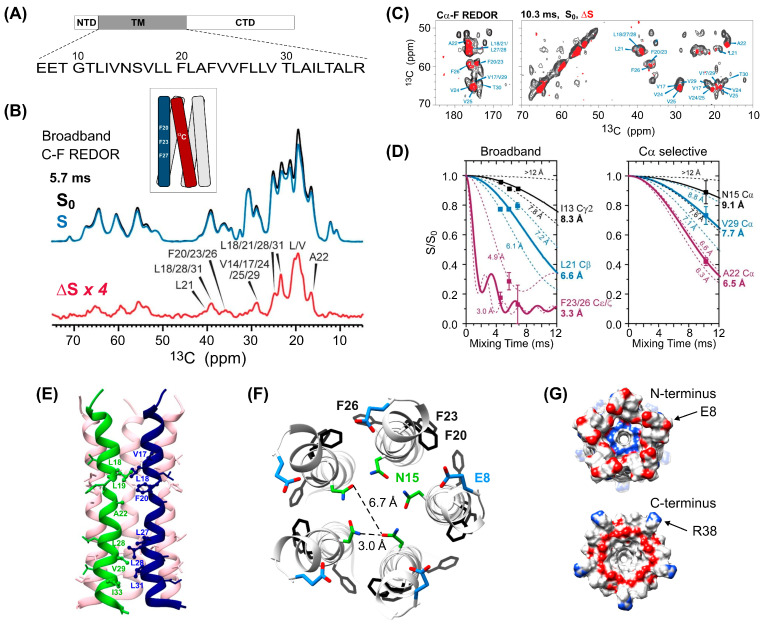
(**A**) Domain architecture of the SARS-CoV-2 envelope protein and the amino acid sequence of its transmembrane segment (ETM). (**B**) Broadband 1D ^13^C–^19^F REDOR full echo (S_o_), dephased (S), and difference (ΔS) spectra of an equimolar mixture of [^13^C]–ETM and [4-^19^F]Phe–ETM reconstituted in ERGIC-mimetic membrane. Schematic representation of the ETM five-helix bundle with ^13^C- and 4-^19^F-Phe-labeled ETM chains is shown as an inset. Signals in the Δ*S* spectrum result from residues, which are in proximity to a fluorinated phenylalanine residue in a neighboring helix. (**C**) The 2D ^13^C_α_{^19^F} REDOR *S_o_* (black) and Δ*S* (red) spectra of the sample in (**B**). (**D**) Representative ^13^C{^19^F} REDOR *S*/*S_o_* curves for broadband and C_α_-selective measurements. The solid lines correspond to the best fit of interhelical ^13^C–^19^F distances for the assigned ^13^C signals. Lower and upper distance bounds are shown as dashed lines. (**E**) Hydrophobic interactions stabilizing the helical interface of the ETM pentamer (PDB code: 7K3G). (**F**) Top view of the pentamer. The N-terminal Glu_8_, the pore facing Asn_15_, and the three phenylalanines are highlighted. (**G**) Surface representation of the pentamer from the N-terminal (top) and the C-terminal (bottom) side. Hydrophobic residues narrow the pore radius to ~2 Å. Adapted with permission from [110].

**Figure 8 ijms-24-13637-f008:**
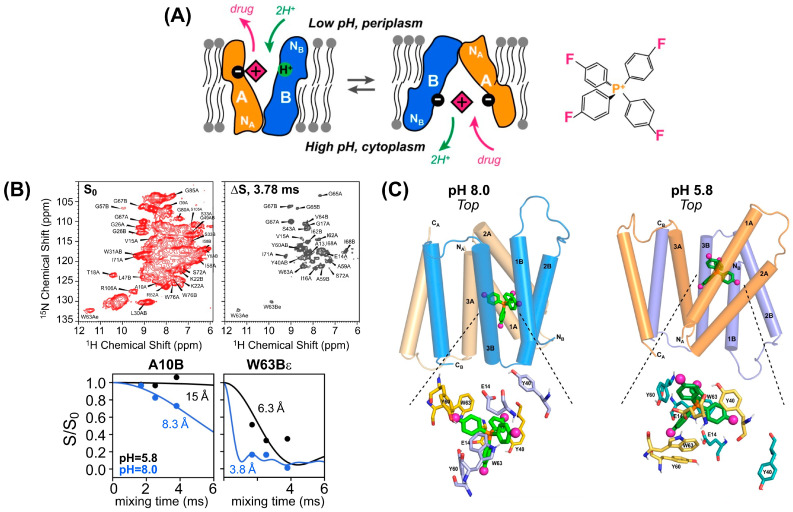
Proton-coupled substrate export in EmrE investigated by REDOR [132,133]. (**A**) Alternating access of EmrE. The asymmetric antiparallel homodimer exchanges between an AB and BA conformation as it converts between the inward- and outward-facing states. The export of toxic polyaromatic cations is coupled to proton import. Substrates and protons bind asymmetrically at a shared binding site. Among the two Glu_14_ residues from the two protomers, one remains accessible to protonation, whereas the other becomes protected. The structure of substrate F_4_-TPP^+^ is shown on the right. (**B**) The ^1^H–^19^F distance measurement between ^15^N-labeled EmrE and ^19^F_4_-TPP^+^ using 2D-hNH-resolved ^1^H{^19^F} REDOR in lipid bilayers. Representative *S_o_* (red) and Δ*S* (black) spectra of the EmrE–substrate complex at pH = 5.8. Assignment is shown for selected peaks. In the Δ*S* spectrum, signal arises only from those H^N^ sites that are proximate to ^19^F. The ^1^H–^19^F REDOR *S*/*S_o_* dephasing curves at pH 5.8 (black) and pH 8.0 (blue) for two representative residues of subunit B (backbone H^N^ of Ala_10_, indole H^N^ of Trp_63_) are shown at the bottom. (**C**) Structure of the EmrE–TPP complex in lipid bilayers at pH 5.8 (PDB: 7JK8) and pH 8.0 (PDB: 7SFQ) determined by solid-state NMR [132,133]. Key residues (E14, Y40, Y60, and W63) of the binding site are shown as sticks. At pH 8.0, the ligand is deeply buried, whereas at pH 5.8 it is more exposed to the periplasmic side. Note that the two protomers swapped conformations between the high- and low-pH complexes.

**Figure 9 ijms-24-13637-f009:**
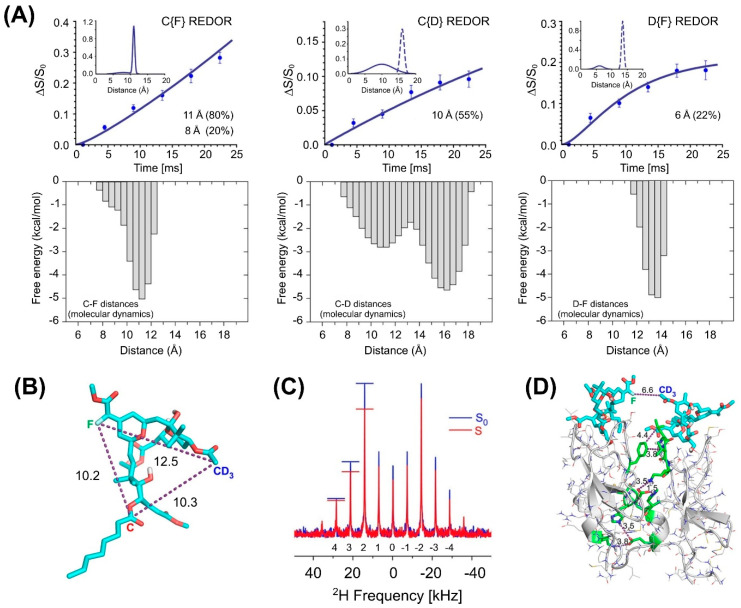
Conformational restraints for PKCδ–C1b-bound bryolog 1 by REDOR [144]. Site-specific ^13^C, ^19^F, and ^2^D labels were introduced at strategically chosen moieties of bryolog 1. (**A**) The ^13^C{^19^F}, ^13^C{^2^D}, and ^2^D{^19^F} REDOR dephasing of the PKCδ–C1b–[^19^F, ^13^C, ^2^D_3_]bryolog 1 complex reconstituted in phospholipid bilayers as a function of dipolar evolution (top). Solid lines correspond to the fitted distribution of distances shown as insets. Free energy as a function of the REDOR-derived distribution of C–F, C–D, and D–F distances in the PKCδ–C1b–bryolog-1–membrane complex calculated by molecular dynamics simulations are shown at the bottom. (**B**) A representative bound bryolog-1 conformer as obtained from REDOR and molecular dynamics simulations. (**C**) The ^2^D{^19^F} REDOR spectra after a dipolar evolution time of 13.44 ms. Differences in side band dephasing indicate orientational preferences of the ^2^D−^19^F internuclear vector. (**D**) Hypothetic dimeric arrangement of two PKCδ–C1b–bryolog 1 complexes fulfilling a 5−7 Å ^2^D−^19^F interligand distance restraint indicated by the presence of a (smaller) population (22%) of distances (panel (**A**), right) based on the REDOR measurements.

**Figure 10 ijms-24-13637-f010:**
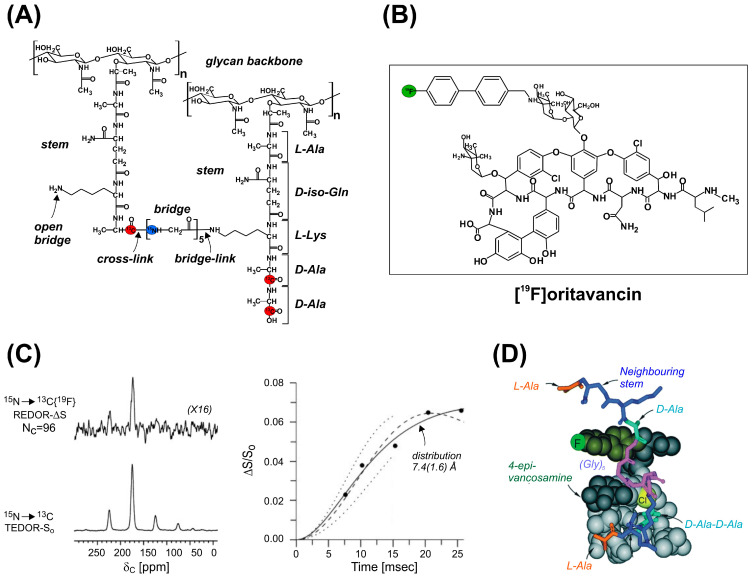
Elucidating drug binding in the cell wall of *S. aureus* [153,154]. (**A**) Chemical structure of the cell wall peptidoglycan of *S. aureus*. The pentapeptide stem having a sequence of L-Ala-D-Glu-L-Lys-D-Ala-D-Ala is attached to N-acetyl muramic acid (NAM) of the disaccharide repeat unit at its first position (L-Ala). The stems are connected by a pentaglycine bridge attached to the ε nitrogen of L-Lys (third position) and a cross-link between the N terminus of the bridge and the D-Ala (fourth position) carbonyl carbon of the adjacent stem. The stem on the right has no cross-link to its D-Ala. Labeled positions upon growing *S. aureus* on media containing D-[1-^13^C]alanine, [^15^N]glycine, and an alanine racemase inhibitor are highlighted. (**B**) Chemical structure of [^19^F]oritavancin, a semi-synthetic vancomycin analogue. (**C**) The ^15^N→^13^C{^19^F} TEDOR-selected REDOR *S_o_* and Δ*S* spectra of *S. aureus* whole cells labeled by D-[1-^13^C]alanine and [^15^N]glycine complexed with [^19^F]oritavancin. The ^15^N→^13^C TEDOR-*S_o_* spectrum (bottom) arises from only ^13^C-labeled D-alanine residues adjacent to a ^15^N-labeled glycine. Following the TEDOR selection, a 96-rotor cycle ^13^C{^19^F} REDOR is applied for the determination of drug–cell wall proximity (top). Magic angle spinning was at 6250 Hz. The ^13^C{^19^F} REDOR dephasing (Δ*S*/*S_o_*) of the cell wall–drug complex as a function of dipolar evolution time is shown on the right. The experimental error was within the size of the solid symbols. The dephasing fitted best to a Gaussian distribution of distances centered at 7.4 Å (width of 1.6 Å) between the ^13^C-labeled carbonyl carbon of the cross-link site and the ^19^F label of the drug. The dotted and dashed lines correspond to the calculated dephasing for single C–F distances of 6.5 Å (dotted, upper), 7.0 Å (dashed), and 7.5 Å (dotted, lower). (**D**) Molecular model of the binding of [^19^F]oritavancin in the cell walls of *S. aureus* generated from several distance measurements including the TEDOR-REDOR 7.4 Å distance.

## Data Availability

Not applicable.

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
