# Peer review of "Three Decades of REDOR in Protein Science: A Solid-State NMR Technique for Distance Measurement and Spectral Editing"

_ijms, 2023, doi:10.3390/ijms241713637_

Round 1
Reviewer 1 Report
In this manuscript for a review article in the International Journal of Molecular Science, Prof. Orsolya Toke reviewed the basic principles and versatility of rotational echo double resonance (REDOR) measurements in biological solid-state NMR. The manuscript is well-written, widely covered, and very useful for all NMR scientists. Further, it mentioned the great contributions and acknowledgments of Prof. J. Shaefer with his co-workers as a pioneer of the REDOR technique and biological applications. I think the manuscript can be accepted after addressing the following minor concerns:
1. Page 24, Line 958-959, ", isogenic Fem-deletion mutants..." ,
Could you insert the full name of "factors essential for methicillin resistance" for "Fem" as follow:
Fem (factors essential for methicillin resistance)-deletion mutants
2. Page 24, Line 978-991 (last part),
The authors introduced more recent works using REDOR in this very nice review. The reviewer thinks that the author should additionally introduce more recent works between 2022-2023.
For example,
Shimizu, K., Mijiddorj, B., Usami, M., Mizoguchi, I., Yoshida, S., Akayama, S., Hamada, Y., Ohyama, A., Usui, K., Kawamura, I., Kawano, R.
De Novo design of a Nanopore for Single Molecule Detection that Incorporates a β-hairpin peptide,
Nature Nanotechnology 2022, 17, 67-75.
Taware, P.P., Jain, M. G., Raran-Kurussi, S., Agarwal, V., Madhu, P.K., Mote, K.R.
Measuring Dipolar Order Parameters in Nondeuterated Proteins Using Solid-State NMR at the Magic-Angle-Spinning Frequency of 100 kHz,
J. Phys. Chem. Lett. 2023, 14, 15, 3627–3635
Xu, H., Yu, H., Chen, Y., Deng, J., Chen, Z., Zhou, P., Lu, J. R., Yang, J., Zhao, Y.
Self-assembly of wide peptide nanoribbons via the formation of nonpolar zippers between β-sheets
Colloids and Surfaces A: Physicochemical and Engineering Aspects, 2023, 659, 130739.
Author Response
Specific Responses
1. Page 24, Line 958-959, ", isogenic Fem-deletion mutants..."
Could you insert the full name of "factors essential for methicillin resistance" for "Fem" as follow:
Fem (factors essential for methicillin resistance)-deletion mutants
Response
I thank the reviewer for his/her comment. I included the full name (currently in pg. 25 of the revised manuscript) and also in the list of abbreviations.
2. Page 24, Line 978-991 (last part),
The authors introduced more recent works using REDOR in this very nice review. The reviewer thinks that the author should additionally introduce more recent works between 2022-2023.
Response
I thank the reviewer for bringing these new works into my attention. I inserted a few sentences about each of them into the revised manuscript with two additional papers from 2022-2023. The following new lines have been inserted:
pg. 9 (ref. to Taware et al., 2023)
“Combination of ε-REDOR [66] and DEDOR (deferred rotational-echo double resonance) [67] allowing the simultaneous measurement of 15N-1H and 13Ca-1Ha dipolar couplings has also been illustrated recently to provide residue specific order parameters for the protein backbone for non-deuterated proteins at high MAS (100 kHz) [68].”
pg. 12 (ref. to Kenyaga et al., 2022)
„A similar experimental strategy has been used in establishing a correlation between membrane insertion depth and fusogenicity in various forms (wild-type vs. mutated, monomer vs. trimer) of the N-terminal 25-residue segment of HIV glycoprotein 41 (gp41) [85] and to probe early stage interactions between the β-amyloid (Aβ) peptide and phospholipid headgroups in synaptic plasma membranes extracted from rat’s brain tissues exploring membrane-associated nucleation processes in fibrillation [86].”
pg. 14 (ref. to Shimizu et al., 2022)
“More recently, 13C{15N} REDOR measurements were used to confirm the conformation of a synthetic peptide with the capability of forming stable nanopores in lipid bilayers with practical applications such as single molecule detection and DNA sequencing [101].”
pg. 17 (refs. to Dregni et al., 2022 and Xu et al., 2023)
“In a more recent study, REDOR and other solid-state NMR measurements were applied to microtubule-associated protein tau [115]. Specifically, 13C-15N distance measurements in cross-β fibrils provided insight into how tau isoforms containing three (3R) or four (4R) microtubule-binding repeats are organized in Alzheimer’s tau tangles. The authors show that the two isoforms are mixed nearly equimolarly and exhibit an isoform-independent preservation of the β-sheet conformation, which, as they hypothesize, may have a role in the predominance of Alzheimer’s disease among neurodegenerative disorders.
Another recent study, along with neutron scattering and atomic force microscopy, employed 13C{15N} REDOR to elucidate the packing mode of an antiparallel β-sheet trimer of a synthetic amphiphatic peptide, Ac-I3VGK-NH2, forming flat nanoribbons [116]. The extensive nonpolar zipper between the β-strands unraveled by the distance measurements exhibited a two-residue shift leading to β-sheet lamination consistent with the observed architecture of the nanoribbon assembly.”
Reviewer 2 Report
The review describes an essential solid-state NMR experiment used by multiple NMR spectroscopists.
I have only a couple of points to improve the review:
1- In Figure 2, the authors show distance restraints extracted from REDOR. It would be beneficial to compare the distances extracted from such pulse sequences in comparison for example, to other famous solution NMR such as NOESY or more sophisticated ones such as eNOE.
2- In the same context: it would be beneficial if the authors could describe how different dipolar-dipolar and CSA tensor can influence the effectiveness of the distances extracted from such a pulse sequence.
3- In Figure 1B: the authors need to highlight why there are two blocks of cross-polarization and clarify the solvent suppression technique used(Mississippi).
4- The text on some figures has insufficient resolution: I would scan the paper for the quality of the text in general.
Author Response
Specific Responses
1- In Figure 2, the authors show distance restraints extracted from REDOR. It would be beneficial to compare the distances extracted from such pulse sequences in comparison for example, to other famous solution NMR such as NOESY or more sophisticated ones such as eNOE.
Response
I thank the reviewer for bringing this into my attention. I fully agree with him/her that such a comparison would be beneficial and the following paragraph has been inserted into the manuscript with additional references (pg. 3-4):
“Unless in conventional solution NMR determination of protein structure, where 1H-1H NOESY (Nuclear Overhauser Effect Spectroscopy) cross-peak volumes at a particular mixing time are translated into qualitative restraints such as a distance range, analysis of dephasing curves as a function of dipolar evolution time in REDOR measurements provides a more exact measure of interatomic distance between the recoupled nuclei, which given the necessary care in experimental design and setup (cf below), can reach an accuracy of ~0.1-0.2 Å. (We note that recently developed exact NOE (eNOE) approaches relying on the extraction of exact NOE rate constants using an iterative protocol between theory and experiment have been shown to reach a similar high accuracy in solution NMR as well [28-29].) Importantly, while NOE-derived interproton distance restraints scale with 1/r6 and has a limit of ~5.5 Å, dephasing in REDOR scales with 1/r3, allowing the determination of longer distances, which depending on the gyromagnetic ratio of the recoupled spin-pair can reach 15-20 Å (Figure 2).”
2- In the same context: it would be beneficial if the authors could describe how different dipolar-dipolar and CSA tensor can influence the effectiveness of the distances extracted from such a pulse sequence.
Response
Similarly, two additional paragraphs have been inserted into the text with additional references discussing the various factors affecting the accuracy of REDOR-derived distances (pg. 4):
“The accuracy of the interatomic distance obtained from REDOR depends on several factors (e.g. resonance offset effects, dipolar couplings other than the desired recoupled interaction, sample and instrument stability, B1 inhomogeneity, appropriate signal-to-noise, proper natural abundance correction). Among these, soon after the development of REDOR was realized that the CSA of the dephasing spin can interfere with the REDOR measurement [30]. To eliminate the detrimental effect of the resonance offset of the dephasing pulses, several phase cycling schemes (xy-4, xy-8, xy-16) were developed [26]. Among them, according to numerical simulations of, for instance the effect of 15N CSA in frequency-selective (FS) 13C-15N REDOR experiments, the xy-4 scheme has been found to be the most adequate [31]. In the case of dephasing nuclei with a larger CSA such as 19F, MAS rates exceeding the CSA of the dephasing spin (~35-40 kHz) are required to eliminate the dependence of the dephasing curve on the orientation of the 19F CSA tensor relative to the recoupled internuclear vector as well as the effect of the large magnitude of CSA on ΔS/So [23,32].
As the determination of longer distances requires longer evolution times, achieving the longest possible T2 of the observed nuclei is of high importance. For this, high MAS and sufficient proton decoupling is necessary. While the So reference experiments compensate for low proton decoupling during the delays between the rf pulses, insufficient decoupling during the dephasing pulses (which at longer evolution times becomes accumulated) may introduce distortions into the observed dephasing [31] highlighting the importance of powerful decoupling schemes. TPPM (two-pulse phase modulation) decoupling [33] of ~80 kHz, for instance, has been shown to be sufficient to minimize 15N-1H interactions in FS 13C-15N REDOR experiments [31]. Additional decoupling schemes and their combinations have been worked out over the years and used successfully for the elimination of the detrimental effect of the dense network of protons in, for instance, Y-detected 1H-X REDOR measurements [34-36], and in 13C-detected FS REDOR measurements aimed at the determination of 1H-13C distances [37]. Of note, as high-power proton decoupling may lead to sample heating, care has to be taken to monitor sample integrity and probe detuning during the course of the experiment. Cycling between the S (dephased) and So (reference) spectra upon collecting the data at each dipolar evolution time point is recommended, which takes care of issues such as instabilities in CP efficiency. More detailed considerations for sample preparation and experimental setup has been given in an excellent summary by Thompson and coworkers [38].”
3- In Figure 1B: the authors need to highlight why there are two blocks of cross-polarization and clarify the solvent suppression technique used(Mississippi).
Response
The caption for Figure 1B has been modified to (pg.3):
„2D 15N-1H resolved 1H-19F REDOR for long-range 1H-19F distance measurement under fast MAS [23]. A REDOR block is inserted before the 1H detection, following a CP-based 2D heteronuclear single-quantum coherence (HSQC) sequence with a Multiple Intense Solvent Suppression Intended for Sensitive Spectroscopic Investigation of Protonated Proteins, Instantly (MISSISSIPPI) solvent suppression block [24]. The pulse sequence employs a 1H to 15N (beginning of the sequence) and a 15N to 1H (before the REDOR block) CP-transfer step. As in (A), the experiment is carried out in two parts. The difference spectrum (ΔS) shows only the amide protons that are proximate to 19F.”
4- The text on some figures has insufficient resolution: I would scan the paper for the quality of the text in general.
Response
We have improved the resolution of the figures and in some cases (e.g. Fig. 3, Fig 8) replaced the text with larger and/or bold letters.